# Evaluation of the US COVID-19 Scenario Modeling Hub for informing pandemic response under uncertainty

Our ability to forecast epidemics far into the future is constrained by the many complexities of disease systems. Realistic longer-term projections may, however, be possible under well-defined scenarios that specify the future state of critical epidemic drivers. Since December 2020, the U.S. COVID-19 Scenario Modeling Hub (SMH) has convened multiple modeling teams to make months ahead projections of SARS-CoV-2 burden, totaling nearly 1.8 million national and state-level projections. Here, we find SMH performance varied widely as a function of both scenario validity and model calibration. We show scenarios remained close to reality for 22 weeks on average before the arrival of unanticipated SARS-CoV-2 variants invalidated key assumptions. An ensemble of participating models that preserved variation between models (using the linear opinion pool method) was consistently more reliable than any single model in periods of valid scenario assumptions, while projection interval coverage was near target levels. SMH projections were used to guide pandemic response, illustrating the value of collaborative hubs for longer-term scenario projections.

Since SARS-CoV-2 was detected in December 2019, there have been numerous disease modeling efforts aiming to inform the pandemic response. These activities have had a variety of goals, including measuring transmissibility, estimating rates of unobserved infections and evaluating control measures[1,2]. Particular attention has been paid to models that attempt to predict the course of the pandemic weeks or months into the future.

These predictive models can, roughly, be divided into two categories: (1) forecasting models that attempt to predict *what will happen* over the future course of the epidemic, encompassing all current knowledge and future uncertainties, and (2) scenario planning models that aim to capture *what would happen if* the future unfolded according to a particular set of circumstances (e.g., intervention policies). While there is no bright line between the two approaches, there are often differences in how they are implemented. Forecasts are typically limited to shorter time horizons, as key drivers of disease dynamics (e.g., human behavior, variant virus emergence) can become highly uncertain at longer horizons. In contrast, scenario projections

often attempt to provide longer term guidance by making explicit assumptions about future changes in those drivers[3], potentially at the expense of predicting *what will happen*. These approaches support decision making in different ways; for instance, forecasts can inform near-term resource allocation and situational awareness[4], while a scenario approach can inform longer-term resource planning and compare potential control strategies[5,6].

Ensembles of independent models consistently outperform individual models in a number of fields[7,8], including infectious disease forecasting[9–12]. Leveraging this multi-model approach, the US COVID-19 Forecast Hub was formed in April 2020, to predict the number of US cases, hospitalizations, and deaths 1-4 weeks into the future[13]. Recognizing that longer term planning scenarios could benefit from a similar multi-model approach[14–16], the US COVID-19 Scenario Modeling Hub (SMH) was formed in December 2020 to produce scenario based projections months into the future.

Between February 2021 and November 2022 SMH produced 16 rounds of projections, 14 of which were released to the public[17] (Round

✉e-mail: ehowerton@psu.edu; viboudc@mail.nih.gov; jlessler@unc.edu

8 was a "practice round", and the emergence of the Omicron variant invalidated Round 10 projections before their release) (Fig. 1). The focus of each round was guided by ongoing discussions with public health partners at the state and federal level and reflected shifting sources of uncertainty in the epidemiology of, and response to, the COVID-19 pandemic. Each round included four scenarios, with early rounds focusing on vaccine availability and use of non-pharmaceutical interventions (NPIs), and later rounds addressing vaccine uptake and the effect of new variants.

In each round, 4-9 modeling teams provided 12 to 52 weeks (depending on the round's goals) of probabilistic projections for each scenario for weekly cases, hospitalizations, and deaths at the state and national level. Projections were aggregated using the linear opinion pool method[18], which preserves variation between model projections[19]. Open calls for projections have yielded participation from thirteen teams overall, with some teams providing projections only for certain rounds or states.

To assess the performance and added value of this large effort we compared SMH projections to real world epidemic trajectories. Whether scenario projections accurately matched those trajectories depends on both how well scenario definitions matched reality, and the calibration of the projections made conditional on those scenarios. Here we attempt to evaluate SMH performance on both criteria (Fig. 2), while acknowledging that there may be complementary evaluations more specific to the many ways SMH projections were used, ranging from informing national vaccine recommendations[5,20] to planning for future COVID-19 surges[21,22]. We present performance results summarized across all SMH rounds, then synthesize these results to discuss performance of SMH projections across different phases of the pandemic.

## Results

### SMH scenarios usually bracketed future epidemic drivers

In each SMH round (except Round 1), four scenarios represented the cells of a 2 × 2 table. Each of the two axes of this table included two

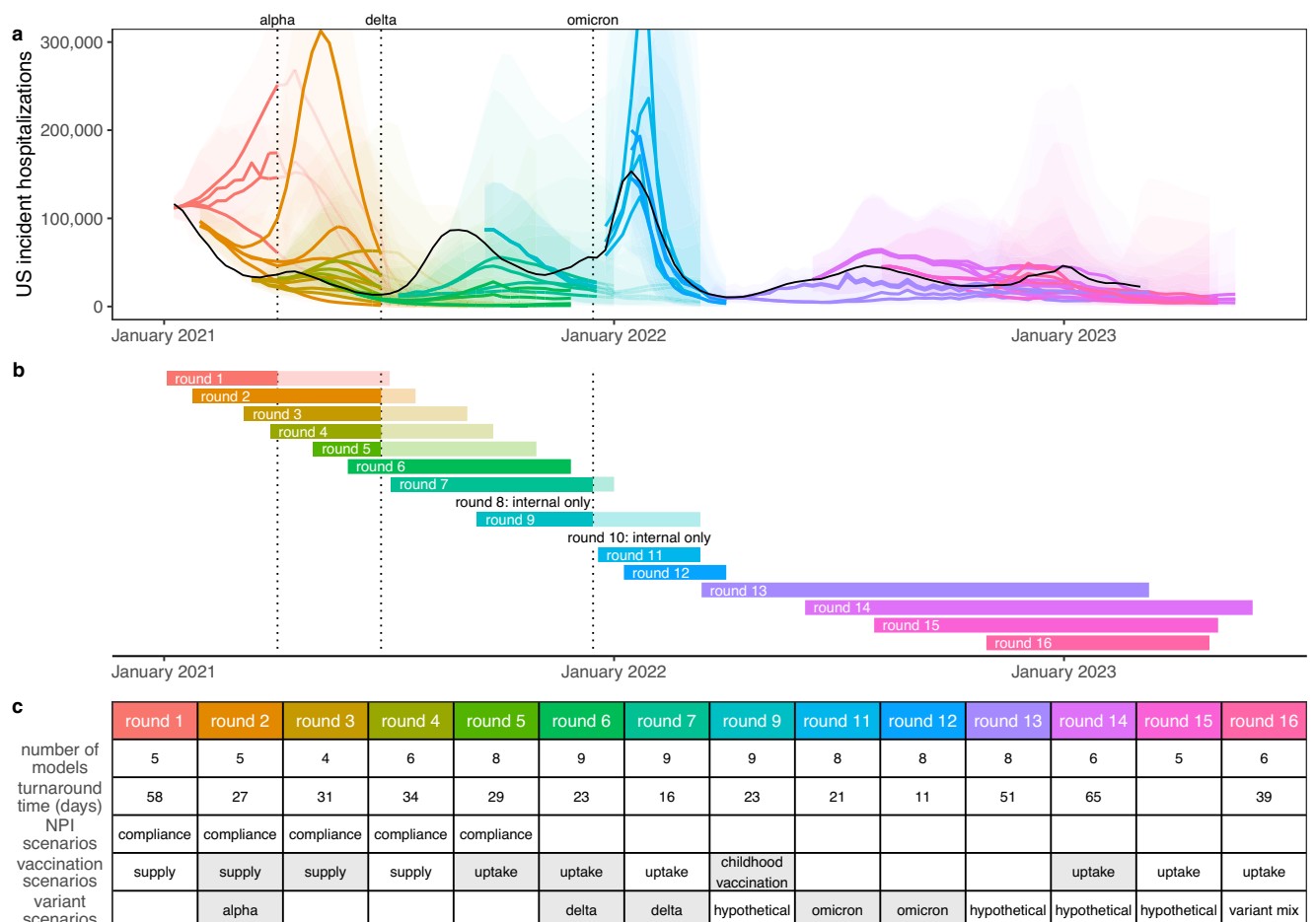

**Fig. 1 | Sixteen rounds of U.S. COVID-19 Scenario Modeling Hub (SMH) projections.** Between February 2021 and November 2022, SMH publicly released fourteen rounds of projections with four scenarios per round. Each round is shown in a different color (internal Rounds 8 and 10 not shown). **a** Median (line) and 95% projection interval (ribbon, the interval within which we expect the observed value to fall with 95% probability, given reality perfectly aligns with the scenario) for U.S. weekly incident hospitalizations for four scenarios per round from the SMH ensemble. Observed weekly U.S. incident hospitalizations are represented by the solid black line. **b** Timing of each round of SMH projections is represented by a projection start date and end date (start and end of bar). In panels (**a**) and (**b**), scenario specifications were invalidated by the emergence of Alpha, Delta, and Omicron variants in rounds that did not anticipate emergence. Variant emergence dates (estimated as the day after which national prevalence exceeded 50%) are represented by dotted vertical lines. **c** For each round, the table specifies the number of participating modeling teams, the turnaround time from finalization of scenarios to publication of projections, and scenario specifications about non-pharmaceutical interventions (NPIs), vaccination, and variant characteristics. Scenario specifications are shaded gray if scenarios "bracketed" the true values in our retrospective analysis (i.e., the true value fell between the two scenario assumptions on that uncertainty axis). Note, in Rounds 11 and 12 both scenario axes specified assumptions about variants, and both are included in the "variant assumptions" cell. Not shown here, the second scenario axis for Round 13 specified assumptions about waning immunity, which bracketed waning estimates from a meta-analysis.

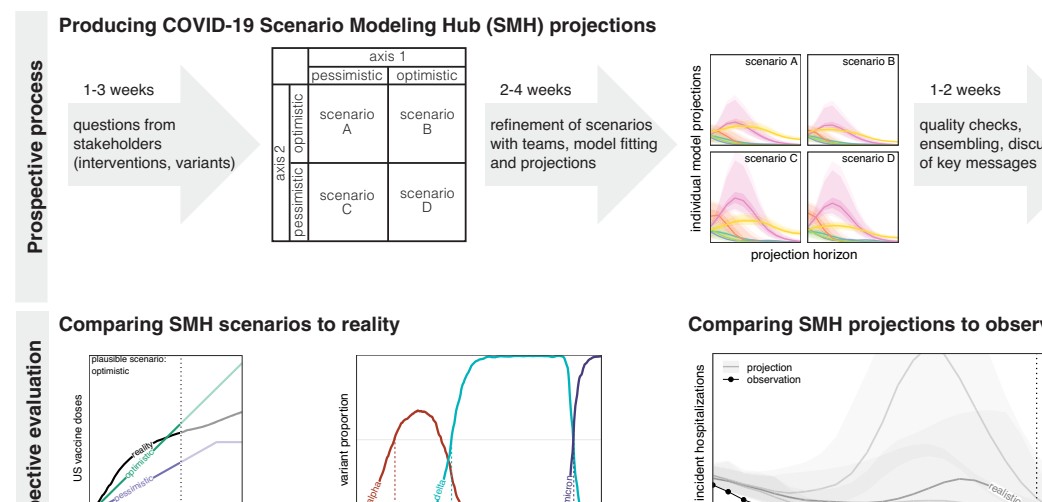

**Fig. 2 | COVID-19 Scenario Modeling Hub (SMH) process.** (top) Prospective SMH process: The SMH coordination team takes input from public health partners on key questions to design scenarios. Scenarios have a 2 × 2 structure (with the exception of Round 1), where two levels are specified along each of two axes of uncertainty or interventions, and all four combinations of these possibilities are considered (scenarios A-D). Scenarios are refined in discussion with modeling teams, after which teams each fit their model and make projections independently. Then, after quality checks, individual model projections are aggregated using linear opinion pool (i.e., probability averaging), and in discussion with the

teams, key messages are determined. A report is shared with public health partners and projections are released on the public SMH website (https://covid19scenariomodelinghub.org). (bottom) Retrospective evaluation: Evaluating the SMH effort involves comparing SMH *scenario assumptions* to reality, and comparing SMH *projections* to observations. Comparing scenarios to reality is used to identify the most plausible scenario-weeks, namely the set of "plausible" scenarios in projection weeks where scenario specifications about variants did not diverge from actual variant prevalence. Horizontal dotted lines represent emergence of an unanticipated variant.

different levels of key sources of uncertainty (e.g., low vs. high variant transmissibility) or intervention (e.g., authorization or not of childhood vaccines) (Fig. 2). Typically, these levels aimed to bracket the future values of important epidemic drivers using information available at the time of scenario design (about, e.g., vaccine hesitancy[23–26], or characteristics of emerging viral variants[27,28]).

We first assessed whether scenario assumptions achieved their goal of bracketing epidemic drivers, as compared to the eventually observed data for those assumptions at the national level (Figs. S2–S4, Table S3). For instance, say one uncertainty axis in a round's scenarios stipulated vaccine coverage would increase up to a low value of 70% and a high value of 80% (depending on the scenario) at the end of the projection period. We say this uncertainty axis "brackets" observations if observed vaccination coverage fell within this range (see Fig. S3 for an example).

Over the 14 publicly released rounds each with two primary axes of uncertainty (i.e., 28 total uncertainty axes), 19 were considered to be evaluable against available observed data (Table 1, see Methods). We succeeded in bracketing at least one axis for the majority of the projection period in 9 of 14 publicly released rounds (14 of 19 evaluable axes). In rounds where one axis specified monthly national vaccine uptake (Rounds 1-4 and 9 for primary series, Rounds 14-15 for boosters), scenarios successfully bracketed observations in 55% of projection weeks (31% Round 1, 100% Round 2, 54% Round 3 and 12% Round 4, 100% for Round 9, 100% Round 14, 38% Round 15, Figs. S2–S5). In 4 other rounds, scenarios specified vaccination coverage at the end of the projection period (Rounds 5-7 for primary series, and 16 for boosters). Assumptions bracketed observed coverage in 2 of these 4 rounds. There were 6 rounds with a scenario axis that attempted to bracket the transmission characteristics (inherent transmissibility or immune escape) of one or more known SARS-CoV-2 variants of concern (Rounds 2, 6, 7, 11, 12, 16). Scenario specifications bracketed most estimates of transmissibility now available in the literature[29,30] (though

one study offers an estimate above the bracketing range for the Delta variant[31]) (Table S3). All rounds including assumptions about variant severity (Rounds 11 and 12) or waning immunity (Round 13) bracketed currently available literature estimates[32–34] (Fig. S6).

The emergence of new variants was a significant challenge in designing scenarios with long term relevance. Changes in the predominantly circulating variant resulted in major divergences from scenario assumptions in 7 of 14 publicly released rounds. Unanticipated variants emerged, on average, 22 weeks into the projection period (median 16 weeks) (Fig. S1), substantially limiting the horizon at which our scenarios remained plausible. This challenge was exacerbated by the lag between when scenarios were defined and when projections were released (5 weeks on average, range 2-10 weeks; Fig. 1), and even led us to cancel release of one SMH round (Round 10) when the Omicron variant emerged. However, in the post Omicron period (Rounds 13-16) SMH scenarios consistently devoted an axis to the emergence of immune escape variants that were deemed consistent with observations, so that projections were considered to remain valid throughout.

**Conditioning on scenario plausibility as a pathway to evaluating projections**

Next we evaluated the performance of SMH projections using prediction interval (PI) coverage and weighted interval score (WIS)[35] (see Methods). PI coverage measures the percent of observations that fall in a prediction interval (so coverage of a 95% PI would ideally be 95%). WIS summarizes calibration across all projection intervals, measuring whether a projection interval captures an observation while penalizing for wider intervals. These standard metrics for evaluating probabilistic forecasts directly compare predictions to observations. In the context of scenario modeling, however, divergences between prediction and observation are the product of two distinct factors: (1) how well the underlying scenario assumptions matched reality (here, scenario

**Table 1 | Scenario bracketing**

| Round | Axis 1 | Axis 2 |
|---|---|---|
| 1 | bracket weekly vaccination coverage in 8 weeks out of 26 weeks (31%) and 8 out of 13 plausible weeks (61%) | *no second bracketing axis* |
| 2 | **bracket weekly vaccination coverage in 26 out of 26 weeks (100%) and 22 out of 22 plausible weeks (100%)** | **bracket variant transmissibility estimates** |
| 3 | **bracket weekly vaccination coverage in 14 out of 26 weeks (54%) and 4 out of 16 plausible weeks (25%)** | *unable to assess NPI scenarios* |
| 4 | bracket weekly vaccination coverage in 3 out of 26 weeks (12%) and 3 out of 13 plausible weeks (23%) | *unable to assess NPI scenarios* |
| 5 | **bracket vaccination coverage at end of projection period** | *unable to assess NPI scenarios* |
| 6 | **bracket vaccination coverage at end of projection period** | **bracket variant transmissibility estimates** |
| 7 | underestimate vaccination coverage in both scenarios | **bracket variant transmissibility estimates** |
| 9 | **bracket weekly vaccination coverage in 19 out of 19 weeks (100%) and 13 out of 13 plausible weeks (100%)** | *no second bracketing axis* |
| 11 | **bracket variant transmissibility estimates** | **bracket variant severity estimates** |
| 12 | **bracket variant transmissibility estimates** | **bracket variant severity estimates** |
| 13 | **bracket immune waning estimates** | *unable to assess immune-escape variant scenarios* |
| 14 | **bracket vaccination coverage in 23 of 23 (100%) evaluated weeks (through March 20, 2023)** | *unable to assess immune-escape variant scenarios* |
| 15 | bracket vaccination coverage in 9 of 24 (38%) evaluated weeks (through March 20, 2023) | *unable to assess immune-escape variant scenarios* |
| 16 | overestimate vaccination coverage in both scenarios | *unable to assess immune-escape variant scenarios* |

For each of two axes per round, bracketing (or not) of reality by U.S. COVID-19 Scenario Modeling Hub (SMH) scenarios. Bold text denotes successful bracketing, and italics text denotes axes where bracketing was not assessed. When vaccination scenarios specified coverage weekly, we considered bracketing in 50% or more of all projection weeks to be bracketing overall. For Round 4, we use coverage of mRNA doses only to determine bracketing, as this makes up almost all of the assumed doses (i.e., we do not consider coverage of Johnson & Johnson). *NPI* non-pharmaceutical intervention. Visualization of scenario assumptions and bracketing are provided in Figs. S2-S6.

plausibility), and (2) how well models would perform in a world where those scenario assumptions are perfectly correct (i.e., model calibration). For instance, if a scenario's definition is highly divergent from real world events, poor predictive accuracy is not necessarily a sign of poor model calibration, and vice versa. Hence, to assess the calibration of SMH models and the ensemble, we need to identify those scenarios and projected weeks where the majority of observed error is likely driven by model miscalibration (i.e., when scenarios are close to reality). We refer to this intersection of scenarios and projected weeks as "plausible scenario-weeks".

To identify the set of plausible scenario-weeks, we first excluded weeks where an emergent variant that was unanticipated in the scenario specifications reached at least 50% prevalence nationally. For evaluation purposes, we considered this to be an invalidation of all remaining scenario-weeks in the round, and thereby removed 79 out of 400 (20%) projection weeks from the plausible set. Then we compared scenario specifications to data on US vaccination coverage and variant characteristics, this time to identify those scenarios that were closest to realized values during non-excluded weeks (see Methods, Table S3 for details). This yielded a total of 292 plausible weeks for calibration analysis (31% of all scenario-weeks), 173 of which (from Rounds 2-4, 13-16) had two plausible scenarios for the same week, which were equally weighted during evaluation.

### SMH ensemble consistently outperformed component and comparator models

An initial question is whether we benefit from aggregating multiple models. To answer this, we assessed the relative calibration of individual models and various ensembling methods across projections from plausible scenario-weeks using overall relative WIS, a metric of performance relative to other models which adjusts for varying projection difficulty across targets (from Cramer et al.[11], see Methods). We assessed variations of two common ensembling techniques: the linear opinion pool (LOP)[18] and the Vincent average[36,37]. The LOP assumes that individual model projections represent different hypotheses about the world and preserves variation between these differing projections[19]. In contrast, the Vincent average assumes that each prediction is an imperfect representation of some common distribution of

interest (like a sample), and accordingly cancels away much of the variation. In practice, as SMH projections started to accumulate, we believed the former assumption better represented the set of SMH models and chose to use a variation of the LOP as our primary approach beginning in Round 4 (where the highest and lowest values are excluded, called the "trimmed-LOP", see Methods). Hereafter, the trimmed-LOP will be referred to as the "SMH ensemble".

We found that the SMH ensemble consistently outperformed component models (Figs. 3c and S48). This ensemble performed better than average, with an overall relative WIS < 1 for all targets, and was the top performer more frequently than any individual model (19 of 42 targets, across 14 rounds with 3 targets per round). It was best or second best 69% of the time (29/42), and in the top 3 performers 93% of the time (39/42). Further, the SMH ensemble partially compensated for the overconfidence of individual models. Across all locations and rounds, overall 95% PI coverage was 79% compared to the ideal 95% for the SMH ensemble versus a median of 40% (interquartile range (IQR) 31-49%) across individual models for incident cases, 80% versus 42% (IQR 31-54%) for incident hospitalizations, and 78% versus 42% (IQR 31-49%) for incident deaths. The trimmed-LOP SMH ensemble also outperformed the two alternative ensembling methods considered (untrimmed-LOP and median Vincent average, Fig. S58).

To assess the added value of SMH, it is important that we compare projections to possible alternatives[38]. In many settings (e.g., weather forecasting) past observations for a similar time of year can be used as a comparator[9,39]. Lacking such historical data for SARS-CoV-2, we chose to compare our projections to two alternate models: (1) a naive model that assumes cases will remain at current levels for the entire projection period with historical variance (the same comparator model used by the COVID-19 Forecast Hub[11]), and (2) a model based on the set of 4-week ahead ensemble predictions from the COVID-19 Forecast Hub (i.e., for any given week predictions from the SMH ensemble were compared to those of the COVID-19 Forecast Hub ensemble made 4-weeks prior). It should be noted that the naive model uses information available at the time of projection, while the 4-week ahead forecast uses more recent observations for most of the projection period.

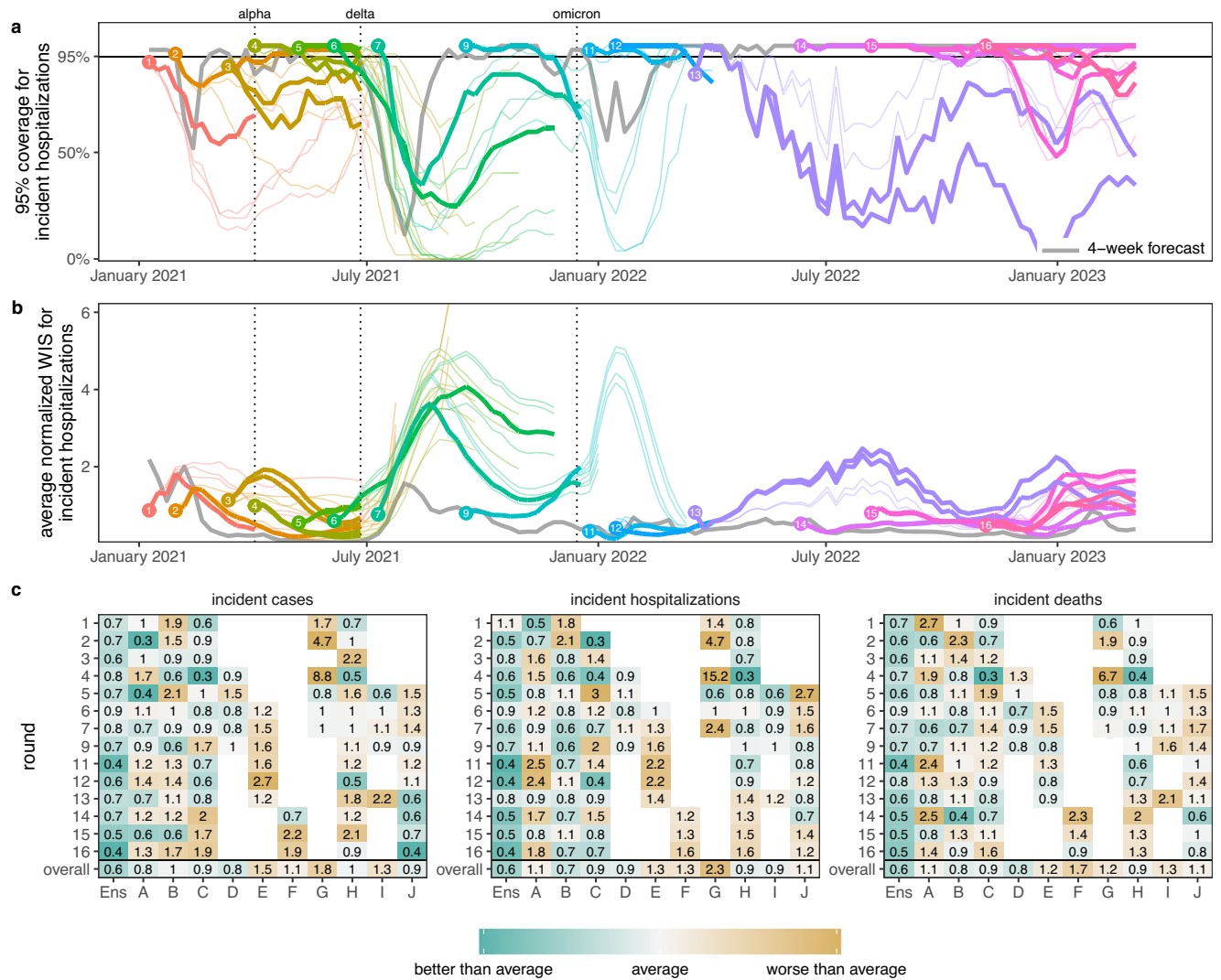

**Fig. 3 | Performance of U.S. COVID-19 Scenario Modeling Hub (SMH) ensemble projections for weekly incident cases, hospitalizations, and deaths. a** Coverage of SMH ensemble 95% projection interval across locations by round and scenario. Ideal coverage of 95% is shown as a horizontal black line. **b** Normalized weighted interval score (WIS) for SMH ensemble by round and scenario. Normalized WIS is calculated by dividing WIS by the standard deviation of WIS across all scenarios and models for a given week, location, target, and round. This yields a scale-free value, and we averaged normalized WIS across all locations for a given projection week and scenario. For (**a**) and (**b**), the round is indicated by color and a number at the start of the projection period. Each scenario is represented by a different line, with plausible scenario-weeks bolded (see Methods). Performance of the

4-wk ahead COVID-19 Forecast Hub ensemble is shown in gray. Vertical dotted lines represent emergence dates of Alpha, Delta, and Omicron variants. Evaluation ended on 10 March 2023, as the source of ground truth observations were no longer produced. **c** Relative WIS comparison of individual models (letters A-I) and SMH ensemble ("Ens") within rounds and overall. A relative WIS of 1 indicates performance equivalent to the "average" model (yellow colors indicate performance worse than average, and greens indicate performance better than average; the color scale is on a log scale and truncated at ±1, representing 2 standard deviations of relative WIS values). See Figs. S46–S47 for 50% and 95% coverage of all targets and see Fig. S22 for comparison of WIS for SMH ensemble to each null comparator.

The SMH ensemble outperformed the naive model across all targets, by 46% for incident cases (relative WIS 0.54, range across rounds 0.14-3.33), 39% for hospitalizations (relative WIS 0.61, range 0.19-1.69) and 58% for deaths (relative WIS 0.42, range 0.07-1.46) (Fig. S22). The SMH ensemble performed worse than the 4-week forecast model overall (relative WIS 1.48, range 0.34-5.79 for cases, 1.41, 0.40-2.85 for hospitalizations, and 2.04, 0.92-3.55 for deaths) (Figs. 3a, b and S22). Occasionally, the SMH ensemble outperformed the 4-week ahead forecast model for cases and hospitalizations, for instance in the highly truncated Round 5 addressing the Alpha variant and the two Omicron rounds (Rounds 11, 12) (Figs. 3a, b, 4, and S14). Some teams that contributed projections to SMH also submitted forecasts to the COVID-19 Forecast Hub, although modeling methodology varied by intended use. In particular, model projections for SMH were conditioned on

specific assumptions that would not necessarily be accounted for in forecasting models.

To better understand the interaction between scenario assumptions and projection performance, we compared average WIS for projections from plausible scenario-weeks with (A) truncated projections from scenarios that were not selected as "most plausible" and (B) all projections, not truncated based on variant emergence. If the ensemble was well calibrated and our selected most plausible scenarios were closest to reality, we would expect projections from plausible scenario-weeks (with truncation) to have the best performance. We found this expectation to be correct in 57% (24/42) of round-target combinations (the other 43% suggesting that SMH ensemble was sometimes "right" for the wrong reasons). Occasionally scenario selection had little effect on performance (e.g., Round 9 and Round 12, Fig. 4). In general, performance for truncated scenarios was

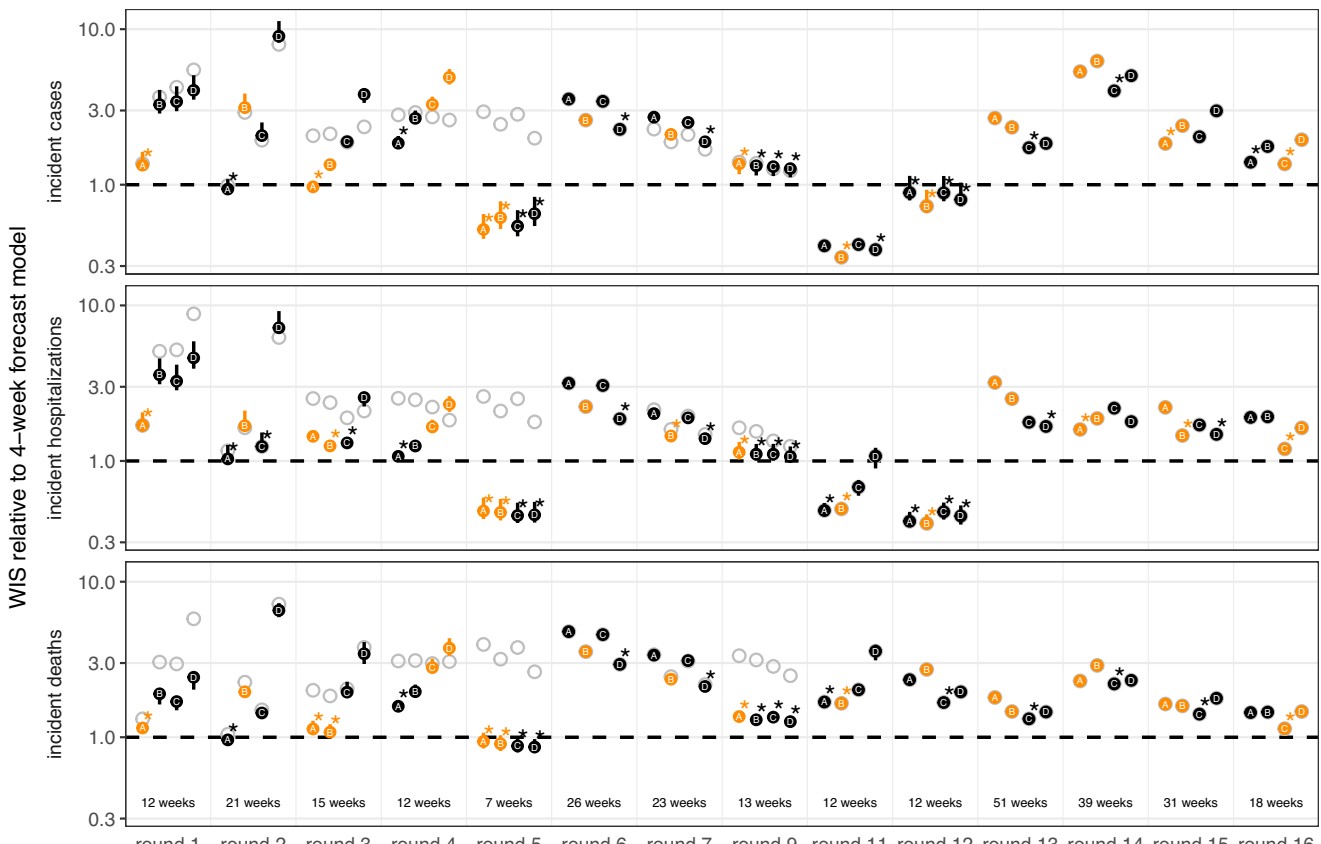

**Fig. 4 | Relative performance of the four U.S. COVID-19 Scenario Modeling Hub (SMH) scenarios (A, B, C, D) across rounds.** Weighted interval score (WIS) for SMH ensemble projections in plausible scenario-weeks relative to the 4-week forecast model (4-week ahead COVID-19 Forecast Hub ensemble). WIS is averaged across all locations and plausible scenario-weeks for a given target, round, and scenario. Scenarios deemed plausible are highlighted in orange (see Methods). The number of plausible weeks included in the average is noted at the bottom of the incident death panel. Results for all weeks are shown with gray open circles for comparison. A WIS ratio of one (dashed line) indicates equal average WIS, or equal performance, between the SMH ensemble and 4-week forecast model. Data are shown as WIS ratio with all weeks included (points), with vertical lines around each point denoting ninety percent (90%) bootstrap intervals. Boostrap intervals are calculated by leaving out WIS for all locations in a given week (over n = 1000 random samples of a single week left out of 7-52 weeks, depending on the number of weeks evaluated in each round). Most bootstrap intervals are very narrow and therefore not easily visible. In each round, the scenario with the lowest WIS ratio is denoted with an asterisk. Any scenario with a 90% bootstrap interval that overlaps the bootstrap interval of the scenario with the lowest WIS ratio is also denoted with an asterisk. WIS ratio is shown on the log scale.

better than if we had not truncated (normalized WIS was the same or lower in 64 of 84 scenario-round-target combinations with truncation), though some of this difference may be attributable to longer projection horizons. Similar conclusions held for 95% PI coverage (Figs. S51–S53).

**While adding value over comparators, SMH projections struggled to anticipate changing disease trends**

Projections may have utility beyond their ability to predict weekly incidence. For instance, projections that predict whether incidence will increase, decrease, or stay the same may be useful even if they are inaccurate in predicting the magnitude of those changes. Based on a method proposed by McDonald et al.[40], we classified projected and observed incident cases, hospitalizations, and deaths in each week and jurisdiction as "increasing", "flat", or "decreasing" using the percent change from two weeks prior (Fig. 5, see Methods).

The median of the SMH ensemble correctly identified the observed trend in 43% of plausible scenario-weeks, comparable to the 4-week forecast model (43%) and better than randomly assigning categories (33%) or assuming continuation of the current trend (34%) (Fig. 5). A classification can also be assessed by the number correctly classified relative to the number predicted (so-called "precision") or the number observed ("recall", see Methods)[41]. Performance on these metrics was similar across targets and classifications, with the exception of correctly anticipating periods of increasing incidence (48% precision and 44% recall for decreasing, 39%/57% for flat, and 45%/24% for increasing, where lower numbers are worse). Although increases were challenging to predict, they have particular public health importance, as these are the periods when interventions or additional resources may be needed. While misses were common, it was relatively rare for the SMH ensemble to predict a decrease when incidence increased (23% of increases) or vice-versa (10% of decreases). A sensitivity analysis based on alternate projection quantiles (other than the median) revealed similar overall performance, though upper quantiles were better at capturing increasing phases (e.g., 95th quantile had 38% precision and 46% recall for increases), at the expense of reduced performance in flat periods (37%/46%, Figs. S36–S37).

**Performance and goals varied over a changing pandemic**

SMH performance varied across different stages of the pandemic. The earliest SMH scenarios (Rounds 1-4) confronted a period of high uncertainty about vaccine supply and the ongoing effect of NPIs. Still, ensemble performance on forecast metrics (WIS, coverage) for plausible scenario-weeks was comparable to average performance across all rounds (Figs. 3 and S42). Of note, the ensemble did not anticipate the increasing and decreasing trends of the Alpha wave well despite

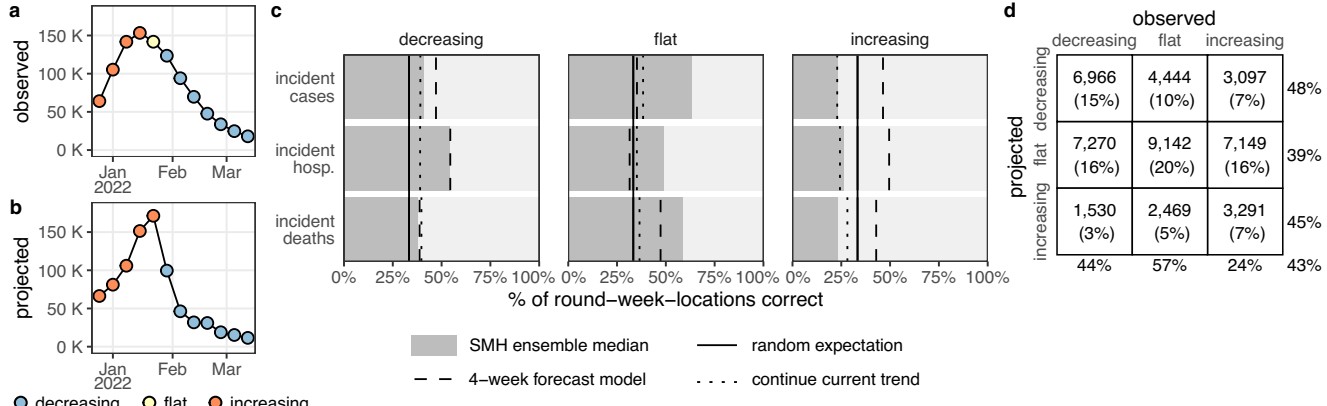

**Fig. 5 | Evaluation of scenario projections to anticipate disease trends.** Illustration of classification of increasing (orange), flat (yellow), and decreasing (blue) trends for observed United States incident hospitalizations (**a**) and U.S. COVID-19 Scenario Modeling Hub (SMH) ensemble projection median for the plausible scenario (**b**) using Round 11, at the start of the Omicron wave. Evaluation of trends across all rounds and locations for plausible scenario-weeks: **c** For decreasing, flat and increasing observations, percent of incident cases, hospitalizations and deaths correctly identified by SMH ensemble projection median (gray), the 4-week forecast model (dashed line), a model that continues current trend (dotted), and the expectation if observations are classified randomly (solid). **d** For decreasing, flat, and increasing observations in plausible scenario-weeks, the number (and percentage) of observations that are classified as decreasing, flat, or increasing by the

SMH ensemble projection median. Totals are calculated across all targets and rounds (meaning that some weeks are included multiple times, and therefore although 33% of observations are in each category, 33% of projections may not be in each category) and weighted by the plausibility of the scenario and week (for rounds with multiple plausible scenarios, this could introduce decimal totals; we rounded values down in these cases). Percentages on the outside show the percent correct for a given observed classification (precision, columns) or projected classification (recall, rows). Projection classifications were also calculated for all scenarios and weeks, regardless of plausibility (Fig. S35), using SMH ensemble projection Q75 (Fig. S36) and SMH ensemble projection Q97.5 (Fig. S37); see supplement for additional stratification of results (by round, Fig. S39; by location, Fig. S40; by projection horizon, Figs. S41–S42; and by variant period, Fig. S43).

including the variant in scenario definitions, with Round 3 missing increases and Round 4 anticipating an overly long and large wave (Fig. S38).

In Rounds 6-7, SMH projections missed the timing and magnitude of the Delta wave, despite scenario assumptions bracketing Delta's transmissibility in both rounds, and vaccine assumptions in Round 6. SMH ensemble performance on forecast metrics was the worst of any period, and trend classification was below par. This miss is likely the result of multiple factors, including unexpectedly rapid waning of vaccine protection, differences in the epidemiology of the Delta variant and earlier viruses (serial interval, intrinsic severity), and changing human behavior in response to the early-summer lull in cases. As more information became available about the Delta variant, SMH projections improved in Round 9 both for forecast metrics and anticipation of epidemic trends (Fig. S45).

During the initial Omicron wave (Rounds 11-12), SMH scenarios anticipated properties of the Omicron variant (all axes bracketed reality), and projections captured weekly trajectories and trends particularly well over the 3 month time horizon. Notably, these were the only rounds without significant truncation where the SMH ensemble outperformed the 4-week ahead forecast for cases and hospitalizations. It is not completely clear why the SMH was able to perform so well during this period. However, scenario designs were well informed by preliminary data from South Africa and heterogeneity in epidemic drivers was low over the projection period (due to high immune escape and relatively stable human behavior), mitigating many of the types of uncertainty that cause particular difficulties for long term epidemic projections.

The first SMH round of the post-Omicron era, Round 13, considered uncertainties about waning immunity and the emergence of a hypothetical immune escape variant. Performance was poor on all statistics and degraded quickly with projection horizon, despite waning assumptions that were consistent with later literature[34]. There was substantial disagreement between models, and projections from some models were highly sensitive to subtle differences in assumptions about the exact trajectory of waning immunity, even when average duration and minimum levels of immune protection were held

constant (Fig. S60). Model disagreement and poor performance may have been further driven by low incidence (hence low information) at the time of calibration.

In contrast, the last three rounds considered here (Rounds 14-16) performed well on forecast metrics over the 18-41 evaluable weeks (key sources of ground truth data became unavailable in March 2023, truncating evaluation). These rounds considered variants with different levels of immune escape and the approval and uptake of bivalent boosters. In these rounds, the SMH ensemble anticipated the occurrence of subsequent waves, was roughly accurate as to their scale, but was less accurate in projecting their timing. Of note, in Round 16 the focus shifted from individual new variants to broad categories of variants with similar levels of immune escape, in an attempt to account for the increasingly complex landscape of SARS-CoV-2 genetic diversity. Still, competition between variants and the resulting dynamics of strain replacement presented challenges for scenario design.

## Discussion

Since December 2020, SMH has convened multiple modeling teams to produce frequent, real-time, probabilistic projections of COVID-19 outcomes over a 3-12 month horizon based on well-defined scenarios. Scenario assumptions bracketed future conditions (where evaluable) the majority of the time, but the relevance of scenarios was frequently truncated by the emergence of unanticipated variants. For projected weeks where scenario assumptions were considered closest to subsequently observed reality, a trimmed linear opinion pool ensemble was far more reliable than any individual model, though anticipating epidemic trends, especially in periods of increasing incidence, remained a challenge. The broad reliability of the ensemble, combined with the alignment of multiple teams on shared questions, helped SMH to become an important source of information for a variety of groups ranging from the media[42] to federal and local public health agencies (e.g.,[1,5,20]).

SMH projections have played an important role in informing the pandemic response to new variants[21,22] and vaccine interventions[5,20]. While the emergence of unanticipated variants presented a challenge to long-term projections, SMH often showed strength in an ability to

anticipate the impact of new variants that were emerging elsewhere in the world. For example, projections from Rounds 6 and 7 sounded an important warning about likely resurgences due to the Delta variant[22], even though performance was poor. Similarly, Round 11 provided important (and ultimately accurate) information about the size and speed of the coming Omicron wave. Notably, SMH projections also provided key information to guide policy recommendations by allowing us to compare different intervention strategies while simultaneously accounting for major uncertainties, such as modeling the emergence of a hypothetical variant. Round 9 addressed potential population-level benefits of childhood vaccination[21], and Rounds 14 and 15 directly informed the decision to recommend bivalent boosters for a wide age range starting September 2022[20]. These public health impacts depended on the timely release (Fig. 1) of projections from scenarios that were both relevant to emergent policy questions and tractable to modeling teams. Consistently fulfilling these goals required frequent meetings and conversations between the coordination team, public health collaborators and modeling teams[43]. This process fostered a vibrant scientific community that has been critical to SMH's success.

Here we have evaluated how SMH scenarios and projections compared to real-world events, with a specific focus on incident cases, hospitalizations and deaths. However, scenario projections may be used in a myriad of ways, and the value of SMH outputs for many of these uses may not directly depend on scenario bracketing or calibration to incident outcomes in plausible scenario-weeks as assessed here. For instance, if the primary goal is to inform a decision about whether or how to implement some intervention, it is the contrast of scenarios with and without that intervention that is important[5,15,16]. Alternatively, one might use the full set of scenarios to allocate resources or inform response plans to potential surges in disease incidence; in this case, we might evaluate how well SMH projections identified states with highest need or the extent to which planning around extremes from pessimistic projections would have led to over- or under-allocation of resources. Our current analysis makes no attempt to directly assess SMH performance for either of these goals (nor to the many other possibilities). Assessing the value added by SMH in these settings would require targeted analyses, and remains an important avenue for future research.

Our analysis of scenario bracketing and model calibration has methodological limitations. We lacked data to evaluate scenario definitions regarding NPIs and certain characteristics of emergent variants, limiting our ability to identify a single most plausible scenario. Teams also had discretion on how to apply vaccination specifications and other scenario assumptions at finer spatial scales; consequently, we did not evaluate scenario plausibility at the state level, although this may have varied substantially. We chose to evaluate SMH projections based on a set of plausible scenario-weeks, but did not account for variability in how closely these plausible weeks matched reality. A complementary approach that may offer better assessments of model calibration would be to re-run scenario projections retrospectively with updated assumptions based on subsequently observed data. Qualitative understanding of the relationship between model assumptions, scenario specifications, and resulting projections can also be useful[44]. In addition, despite the fact that forecast models projecting over a shorter time horizon can use more recent information, post-hoc selection of plausible scenario weeks has the potential to "tip the scales" of evaluation in favor of scenario projections, as forecast models are not given the opportunity to project under multiple scenarios. There also remain many open questions about the predictability of infectious disease systems, such as the relative benefits of recent calibration data (which would benefit forecast models) versus knowledge of key drivers of disease dynamics (which would benefit scenario projection models that consider multiple possibilities). Lastly, without a good comparator model it is hard to evaluate

the added value of the SMH projections, and lack of historic data and the nature of planning scenarios makes design of such a comparator difficult.

The scenario approach is an attempt to provide useful projections in the face of the many complexities that make predicting epidemics difficult. One of the most important complexities is the multiple, interacting drivers of disease dynamics that are themselves difficult to predict, such as ever evolving pathogen characteristics and human behavior. Although the scenario approach allows us to provide projections despite these complexities, only a subset of possible futures are explored. Therefore, it is essential to design scenarios that are useful – narrowing in on the possible futures that will best inform present actions. The fast timescale and multi-wave nature of infectious disease outbreaks often means we have little time to deeply consider both scenario design and model implementation in real time, but it allows us to learn about the system and refine our approaches to scenario design and epidemic modeling more quickly than is possible in other systems (e.g., climate[45]).

Since its inception, SMH has disseminated nearly 1.8 million unique projections, making it one of the largest multi-team infectious disease scenario modeling efforts to date (other notable efforts include multi-model estimation of vaccination impact[46–48], planning for future influenza pandemics[49], and COVID-19 response in South Africa[50] and the UK[44]). The SMH process, which uses the power of multi-model ensembles and strategic selection of future scenarios to manage uncertainty[15], has already been replicated in other settings[51] and for other pathogens[52]. Looking to the future, the lessons learned and the emerging shared hub infrastructure[53] can help to provide a more effective, coordinated, and timely response to new pandemic threats and improve mitigation of endemic pathogens. It will be advantageous to launch multi-model efforts for scenario planning, forecasting, and inference in the early stages of future pandemics, when the most critical, time-sensitive decisions need to be made and uncertainty is high. To do this effectively, we can build on the SMH and other efforts from the COVID-19 response by continuing "peace time" research into how to better collect and use data, construct scenarios, build models, and ensemble results. As part of an evidence-based pandemic response, scenario modeling efforts like SMH can support decision making through improved predictive performance of multi-model ensembles and well-defined shared scenarios.

## Methods
### Overview of evaluation approaches for scenario projections
When evaluating the distance between a scenario projection and an observation, there are two potential factors at play: (1) the scenario assumptions may not match reality (e.g., scenario-specified vaccine uptake may underestimate realized uptake), and (2) if there were to be alignment between the scenario specifications and reality, model predictions may be imperfect due to miscalibration. The difference between projections and observations is a complex combination of both sources of disagreement, and importantly, observing projections that are close to observations does not necessarily imply projections are well-calibrated (i.e., for scenarios very far from reality, we might expect projections to deviate from observations). To address both components, we evaluated the plausibility of COVID-19 Scenario Modeling Hub (SMH) scenarios and the performance of SMH projections (ensemble and component models). A similar approach has been proposed by Hausfather et al.[45]. Below, we describe in turn the component models contributing to SMH, the construction of the ensemble, the evaluation of scenario assumptions, and our approaches to estimating model calibration and SMH performance.

### Elicitation methods and models submitting projections to SMH
SMH advertised new rounds of scenario projections across various modeling channels, using an open call to elicit projections from

independent modeling teams. Scenario specifications were designed in collaboration with public health partners and modeling teams, and final specifications were published on a public GitHub repository (https://github.com/midas-network/covid19-scenario-modeling-hub). Teams then submitted projections to this same repository. For additional discussion about the philosophy and history of SMH, as well as details about SMH process, see Loo et al.[43]

Over the course of the first sixteen rounds of SMH, thirteen independent models submitted projections, with most submitting to multiple rounds. Of participating models, prior experience in public health modeling varied substantially, ranging from teams with newly built models to address the COVID-19 pandemic and those with long-established relationships with local, state, and national public health agencies. The majority of submitting models were mechanistic compartmental models, though there was one semi-mechanistic model and two agent-based models. Some models were calibrated to, and made projections at, the county level, whereas others were calibrated to and made projections at the state level; many, but not all, had age structure. We have provided an overview of each model in Table S1. As models changed each round to accommodate different scenarios and adapt to the evolving pandemic context, we chose not to focus here on model-specific differences (in structure, parameters, or performance). For more information on round-specific implementations, we direct readers to other publications with details[5,22].

### Inclusion criteria and projections used for evaluation

Our analysis included state- and national-level projections of weekly incident cases, hospitalizations, and deaths from individual models and various ensembles for fourteen of the first sixteen rounds of SMH (Rounds 8 and 10 were not released publicly, and therefore are not included; see also Table S2 for a list of jurisdictions included). Each round included projections from between 4 and 9 individual models as well as ensembles. For a given round, modeling teams submitted projections for all weeks of the projection period, all targets (i.e., incident or cumulative cases, hospitalizations, and deaths), all four scenarios, and at least one location (i.e., states, territories, and national). Here, we evaluated only individual models that provided national projections in addition to state-level projections (i.e., excluding individual models that did not submit a national projection, though projections from these models are still included in the state-level ensembles that were evaluated). Submitted projections that did not comply with SMH conditions (e.g., for quantifying uncertainty or defining targets) were also excluded (0.8% of all submitted projections). Detailed description of exclusions can be found in Table S2.

### Probabilistic projections and aggregation approaches

Modeling teams submitted probabilistic projections for each target via 23 quantiles (e.g., teams provided projected weekly incident cases for Q1, Q2.5, Q5, Q10, Q20, ..., Q80, Q90, Q95, Q97.5, and Q99). We evaluated 3 methods for aggregating projections: untrimmed-LOP, trimmed-LOP (variations of probability averaging or linear opinion pool[18], LOP), and median-Vincent (variation of quantile or Vincent averaging[36,37] which is also used by other hubs[11]).

The untrimmed-LOP is calculated by taking an equally weighted average of cumulative probabilities across individual models at a single value. Because teams submitted projections for fixed quantiles, we used linear interpolation between these value-quantile pairs to ensure that all model projections were defined for the same values. We assumed that all projected cumulative probabilities jump to 0 and 1 outside of the defined value-quantile pairs (i.e., Q1-Q99). In other words, for a projection defined by cumulative distribution $F(x)$ with quantile function $F^{-1}(x)$, we assume that $F(x) = 0$ for all $x < F^{-1}(0.01)$ and $F(x) = 1$ for all $x > F^{-1}(0.99)$.

The trimmed-LOP uses exterior cumulative distribution function (CDF) trimming[54] of the two outermost values to reduce the variance

of the aggregate, compared to the untrimmed-LOP (i.e., the prediction intervals are narrower). To implement this method, we follow the same procedure as the untrimmed-LOP, but instead of using an equally-weighted average, we exclude the highest and lowest quantiles at a given value and equally weight all remaining values in the average. Under this trimming method, the exclusions at different values may be from different teams.

The median-Vincent aggregate is calculated by taking the median value for each specified quantile. These methods were implemented using the CombineDistributions package[19] for the R statistical software[55].

### Scenario design and plausibility

Projections in each SMH round were made for 4 distinct scenarios that detailed potential interventions, changes in behavior, or epidemiologic situations (Fig. 1). Scenario design was guided by one or more primary purposes[56], which were often informed by public health partners and our hypotheses about the most important uncertainties at the time. SMH scenarios were designed approximately one month before projections were submitted, and therefore 4-13 months before the end of the projection period, depending on the round's projection horizon. Scenario assumptions, especially those about vaccine efficacy or properties of emerging viral variants, were based on the best data and estimates available at the time of scenario design (these were often highly uncertain). Here, our purpose was to evaluate SMH scenario assumptions using the best data and estimates currently available, after the projection period had passed. We assessed SMH scenarios from two perspectives:

1. based on their *prospective* purpose: we identified whether scenarios "bracketed" reality along each uncertainty axis (i.e., one axis of the 2 × 2 table defining scenarios, based on one key source of uncertainty for the round). Scenarios in most SMH rounds were designed to bracket true values of key epidemic drivers (though the true value was not known at the time of scenario design). In other words, along each uncertainty axis in an SMH round, scenarios specified two levels along this axis (e.g., "optimistic" and "pessimistic" assumptions). Here we tested whether the realized value fell between those two assumptions (if so, we called this "bracketing").

2. for *retrospective* evaluation of calibration: we identified the set of plausible scenario-weeks for each round. One of our primary goals in this analysis was to assess and compare the calibration of different approaches (e.g., individual models, SMH ensemble, comparator models). To assess this in the most direct way possible, we chose scenarios and projection weeks that were close to what actually happened (i.e., we isolated error due to calibration by minimizing deviation between scenarios and reality; see *Overview of evaluation approaches for scenario projections* for details).

An "evaluable" scenario axis was defined as an axis for which assumptions could be confronted with subsequently observed data on epidemic drivers; in some instances, we could not find relevant data and the axis was not considered evaluable (e.g., NPI, see below). To evaluate scenario assumptions, we used external data sources and literature (Table S3). Due to differences across these sources, we validated each type of scenario assumption differently (vaccination, NPI, and variant characteristics; Fig. 2), as described in detail below and in Table S3. Vaccine specifications and realized coverage are shown in Figs. S2-S5, while details of our round-by-round evaluation are provided below.

Rounds 1-4 concentrated on the early roll-out of the vaccine in the US and compliance with NPIs. To evaluate our vaccine assumptions in these rounds, we used data on reported uptake from the US Centers for Disease Control and Prevention database[57]. For these rounds,

scenarios prescribed monthly national coverage (state-specific uptake was intentionally left to the discretion of the modeling teams), so we only used national uptake to evaluate the plausibility of each vaccination scenario (Fig. S2). In these scenarios, "bracketing" was defined as reality falling between cumulative coverage in optimistic and pessimistic scenarios for 50% or more of all projection weeks. The "plausible" scenario was that scenario with the smallest absolute difference between cumulative coverage in the final projection week (or in cases of variant emergence, the last week of projections before emergence; details below) and the observed cumulative coverage. We also considered choosing the plausible scenario via the cumulative difference between observed and scenario-specified coverage over the entire projection period; this always led to selecting the same scenario as plausible.

When scenarios specified a coverage threshold, we compared assumptions with the reported fraction of people vaccinated at the end of the projection period. For instance, in Round 2 scenario C and D, we stipulated that coverage would not exceed 50% in any priority group, but reported vaccination exceeded this threshold. In Rounds 3-4, the prescribed thresholds were not exceeded during the truncated projection period.

By Round 5 (May 2021), vaccine uptake had started to saturate. Accordingly, in Rounds 5-7, vaccine assumptions were based on high and low saturation thresholds that should not be exceeded for the duration of the projection period, rather than monthly uptake curves. For these rounds, we evaluated which of the prescribed thresholds was closest to the reported cumulative coverage at the end of the projection period (Fig. S3). Later rounds took similar approaches to specifying uptake of childhood vaccination (Round 9) and bivalent boosters (Round 14-16). Rounds 9 (Fig. S4), and 14-15 (Fig. S5) specified weekly coverage and Round 16 specified a coverage threshold; we followed similar approaches in evaluating these scenarios.

For vaccine efficacy assumptions, we consulted population-level studies conducted during the period of the most prevalent variant during that round (Table S3). Similarly, for scenarios about emerging viral variants (regarding transmissibility increases, immune escape, and severity) and waning immunity, we used values from the literature as a ground truth for these scenario assumptions. We identified the most realistic scenario as that with the assumptions closest to the literature value (or average of literature values if multiple were available, Table S3).

Rounds 1-4 included assumptions about NPIs. We could not identify a good source of information on the efficacy of and compliance to NPIs that would match the specificity prescribed in the scenarios (despite the availability of mobility and policy data, e.g., Hallas et al.[58]). Rounds 13-15 included assumptions about immune escape and severity of hypothetical variants that may have circulated in the post-Omicron era. Round 16 considered broad variant categories based on similar levels of immune escape, in response to the increasing genetic diversity of SARS-CoV-2 viruses circulating in fall 2022. There were no data available for evaluation of immune escape assumptions after the initial Omicron BA.1 wave. As such, NPI scenarios in Rounds 1-4 and immune escape variant scenarios in Rounds 13-16 were not "evaluable" for bracketing analyses, and therefore we considered all scenarios realistic in these cases. Overall, across 14 publicly released rounds, we identified a single most realistic scenario in 7 rounds, and two most realistic scenarios in the other 7.

Finally, in some rounds, a new viral variant emerged during the projection period that was not specified in the scenarios for that round. We considered this emergence to be an invalidation of scenario assumptions, and removed these weeks from the set of plausible scenario-weeks. Specifically, emergence was defined as the week after prevalence exceeded 50% nationally according to outbreak.info variant reports[59–61], accessed via outbreak.info R client[62]. Accordingly, the Alpha variant (not anticipated in Round 1 scenarios) emerged on 3 April 2021, the Delta variant (not anticipated in Rounds 2-5) emerged on 26 June 2021, and the Omicron variant (not anticipated in Round 9) emerged on 25 December 2021.

## Comparator models

To assess the added value of SMH projections against plausible alternative sources of information, we also assessed comparator models or other benchmarks. Comparator models based on historical data were not available here (e.g., there was no prior observation of COVID-19 in February in the US when we projected February 2021). There are many potential alternatives, and here we used three comparative models: naive, 4-week forecast, and trend-continuation.

The baseline "naive" model was generated by carrying recent observations forward, with variance based on historical patterns (Figs. S13–S15). We used the 4-week ahead "baseline" model forecast from the COVID-19 Forecast Hub[11] for the first week of the projection period as the naive model, and assumed this projection held for the duration of the projection period (i.e., this forecast was the "naive" projection for all weeks during the projection period). Because the COVID-19 Forecast Hub collects daily forecasts for hospitalizations, we drew 1000 random samples from each daily distribution in a given week and summed those samples to obtain a prediction for weekly hospitalizations. The naive model is flat and has relatively large prediction intervals in some instances.

As a forecast-based comparator, we used the COVID-19 Forecast Hub "COVIDhub-4_week_ensemble" ensemble model (Figs. S7–S9). This model includes forecasts (made every week) from multiple component models (e.g., on average 41 component models between January and October 2021[11]). We obtained weekly hospitalization forecasts from the daily forecasts of the COVID-19 Forecast Hub using the same method as the naive model. This 4-week forecast model is particularly skilled at death forecasts[11]; however, in practice, there is a mismatch in timing between when these forecasts were made and when SMH projections were made. For most SMH projection weeks, forecasts from this model would not yet be available (i.e., projection horizons more than 4 weeks into the future); yet, for the first 4 weeks of the SMH projection period, SMH projections may have access to more recent data. It should also be noted that the team running the COVID-19 Forecast Hub has flagged the 4-week ahead predictions of cases and hospitalizations as unreliable[63]. Further, SMH may be given an "advantage" by the post-hoc selection of plausible scenario-weeks based on the validity of scenario assumptions.

Finally, the trend-continuation model was based on a statistical generalized additive model (Figs. S10–S12). The model was fit to the square root of the 14-day moving average with cubic spline terms for time, and was fit separately for each location. We considered inclusion of seasonal terms, but there were not enough historic data to meaningfully estimate any seasonality. For each round, we used only one year of data to fit the model, and projected forward for the duration of the projection period. The SMH ensemble consistently outperformed this alternative comparator model (see Figs. S16–S21).

## Projection performance

Prediction performance is typically based on a measure of distance between projections and "ground truth" observations. We used the Johns Hopkins CSSE dataset[64] as a source of ground truth data on reported COVID-19 cases and deaths, and U.S. Health and Human Services Protect Public Data Hub[65] as a source of reported COVID-19 hospitalizations. These sources were also used for calibration of the component models. CSSE data were only produced through 4 March 2023, so our evaluation of Rounds 13-16 ended at this date (1 week before the end of the 52 week projection period in Round 13, 11 weeks before the end of the 52 week projection period in Round 14, 9 weeks before the end of the 40 week projection period in Round 15, and 8 weeks before the end of the 26 week projection period in Round 16).

We used two metrics to measure performance of probabilistic projections, both common in the evaluation of infectious disease predictions. To define these metrics, let $F$ be the projection of interest (approximated by a set of 23 quantile-value pairs) and $o$ be the corresponding observed value. The "$\alpha$% prediction interval" is the interval within which we expect the observed value to fall with $\alpha$% probability, given reality perfectly aligns with the specified scenario.

1. **Ninety-five percent (95%) coverage** measures the percent of projections for which the observation falls within the 95% projection interval. In other words, 95% coverage is calculated as

$$C_{95\%}(F,o) = \frac{1}{N}\sum_{i=1}^{N} 1\left(F^{-1}(0.025) \le o \le F^{-1}(0.975)\right) \quad (1)$$

where $1(\cdot)$ is the indicator function, i.e., $1(F^{-1}(0.025) \le o \le F^{-1}(0.975)) = 1$ if the observation falls between the values corresponding to Q2.5 and Q97.5, and is 0 otherwise. We calculated coverage over multiple locations for a given week (i.e., $i = 1...N$ for $N$ locations), or across all weeks and locations.

2. **Weighted interval score (WIS)** measures the extent to which a projection captures an observation, and penalizes for wider prediction intervals[35]. First, given a projection interval (with uncertainty level $\alpha$) defined by upper and lower bounds, $u = F^{-1}\left(1 - \frac{\alpha}{2}\right)$ and $l = F^{-1}\left(\frac{\alpha}{2}\right)$, the interval score is calculated as

$$IS_\alpha(F,o) = (u - l) + \frac{2}{\alpha}(l - o)1(o < l) + \frac{2}{\alpha}(o - u)1(u < o) \quad (2)$$

where again, $1(\cdot)$ is the indicator function. The first term of $IS_\alpha$ represents the width of the prediction interval, and the second two terms are penalties for over- and under-prediction, respectively. Then, using weights that approximate the continuous rank probability score[66], the weighted interval score is calculated as

$$WIS(F,o) = \frac{1}{K+1/2}\left(\frac{1}{2}|o - F^{-1}(0.5)| + \sum_{i=1}^{K}\frac{\alpha_K}{2}IS_\alpha\right) \quad (3)$$

Each projection is defined by 23 quantiles comprising 11 intervals (plus the median), which we used for the calculation of WIS (i.e., we calculated $IS_\alpha$ for $\alpha = 0.02, 0.05, 0.1, 0.2,...,0.8, 0.9$ and $K = 11$). It is worth noting that these metrics do not account for measurement error in the observations.

WIS values are on the scale of the observations, and therefore comparison of WIS across different locations or phases of the pandemic is not straightforward (e.g., the scale of case counts is very different between New York and Vermont). For this reason, we generated multiple variations of WIS metrics to account for variation in the magnitude of observations. First, for average normalized WIS (Fig. 3b), we calculated the standard deviation of WIS, $\sigma_{s,w,t,r}$, across all scenarios and models for a given week, location, target, and round and divided the WIS by this standard deviation (i.e., $WIS/\sigma_{s,w,t,r}$). Doing so accounts for the scale of that week, target, and round, a procedure implemented in analyses of climate projections[67]. Then, we averaged normalized WIS values across strata of interest (e.g., across all locations, or all locations and weeks). Other standardization approaches that compute WIS on a log scale have been proposed[68], though may not be as well suited for our analysis which focuses on planning and decision making.

An alternative rescaling introduced by Cramer et al.[11], relative WIS, compares the performance of a set of projections to an "average" projection. This metric is designed to compare performance across predictions from varying pandemic phases. The relative WIS for model $i$ is based on pairwise comparisons (to all other models, $j$) of average WIS. We calculated the average WIS across all projections in common between model $i$ and model $j$, where $WIS(i)$ and $WIS(j)$ are the average WIS of these projections (either in one round, or across all rounds for

"overall") for model $i$ and model $j$, respectively. Then, relative WIS is the geometric average of the ratio, or

$$\text{relative WIS} = \left(\prod_{j=1}^{N}\frac{WIS(i)}{WIS(j)}\right)^{1/N} \quad (4)$$

When comparing only two models that have made projections for all the same targets, weeks, locations, rounds, etc. the relative WIS is equivalent to a simpler metric, the ratio of average WIS for each model (i.e., $\frac{WIS(i)}{WIS(j)}$). We used this metric to compare each scenario from SMH ensemble to the 4-week forecast model (Fig. 4). For this scenario comparison, we provided bootstrap intervals by recalculating the ratio with an entire week of projections excluded (all locations, scenarios). We repeated this for all weeks, and randomly drew from these 1000 times. From these draws we calculated the 5th and 95th quantiles to derive the 90% bootstrap interval, and we assumed performance is significantly better for one scenario over the others if the 90% bootstrap intervals do not overlap. We also used this metric to compare the ensemble projections to each of the comparative models (Fig. S22).

## Trend classification

In addition to traditional forecast evaluation metrics, we assessed the extent to which SMH projections predict the qualitative shape of incident trajectories (whether trends will increase or decrease). We modified a method from McDonald et al.[40] to classify observations and projections as "increasing", "flat" or "decreasing". First, we calculated the percent change in observed incident trajectories on a two week lag (i.e., $\log(o_T + 1) - \log(o_{T-2} + 1)$ for each state and target). We took the distribution of percent change values across all locations for a given target and set the threshold for a decrease or increase assuming that 33% of observations will be flat (Fig. S23). Based on this approach, decreases were defined as those weeks with a percent change value below −23% for incident cases, −17% for incident hospitalizations, and −27% for incident deaths, respectively. Increases have a percent change value above 14%, 11%, 17%, respectively. See Fig. S34 for classification results with a one week lag and different assumptions about the percent of observations that are flat.

Then, to classify trends in projections, we again calculated the percent change on a two week lag of the projected median (we also consider the 75th and 95th quantiles because our aggregation method is known to generate a flat median when asynchrony between component models is high). For the first two projection weeks of each round, we calculated the percent change relative to the observations one and two weeks prior (as there are no projections to use for reference in the week prior, and two weeks prior, projection start date). We applied the same thresholds from the observations to classify a projection, and compared this classification to the observed classification. This method accounts for instances when SMH projections anticipate a change in trajectory but not the magnitude of that change (see Fig. S44), and it does not account for instances when SMH projections anticipate a change but miss the timing of that change (this occurred to some extent in Rounds 6 and 7, Delta variant wave). See Figs. S24–S33 for classifications of all observations and projections.

We assessed how well SMH projections captured incident trends using precision and recall, two common metrics in evaluating classification tasks with three classes: "increasing", "flat", and "decreasing"[41]. To calculate these metrics, we grouped all projections by the projected and the observed trend (as in Fig. 5d). Let $N_{po}$ be the number of projections classified by SMH as trend $p$ (rows of Fig. 5d) and the corresponding observation was trend $o$ (columns of Fig. 5d). All possible combinations are provided in Table 2. Then, for class $c$ (either decreasing, flat, or increasing),

**Table 2 | Number of projections classified as decreasing, flat, or increasing based on both the projected and the observed classes**

| | | observed | | |
|---|---|---|---|---|
| | | decreasing | flat | increasing |
| projected | decreasing | $N_{DD}$ | $N_{DF}$ | $N_{DI}$ |
| | flat | $N_{FD}$ | $N_{FF}$ | $N_{FI}$ |
| | increasing | $N_{ID}$ | $N_{IF}$ | $N_{II}$ |

Correctly classified projections fall on the diagonal, i.e., $N_{DD}$, $N_{FF}$, and $N_{II}$.

1. *precision* is the fraction of projections correctly classified as $c$, out of the total number of projections classified as $c$, or

$$\text{precision}_c = \frac{N_{cc}}{\sum_{j=1}^{3} N_{cj}} \qquad (5)$$

For example, the precision of increasing trends is the number of correctly classified increases ($N_{II}$) divided by the total number of projections classified as increasing ($N_{ID} + N_{IF} + N_{II}$).

2. *recall* is the fraction of projections correctly classified as $c$, out of the total number of projections observed as $c$, or

$$\text{recall}_c = \frac{N_{cc}}{\sum_{j=1}^{3} N_{jc}} \qquad (6)$$

For example, the recall of increasing trends is the number of correctly classified increases ($N_{II}$) divided by the total number of observations that increased ($N_{DI} + N_{FI} + N_{II}$).

In some instances, we provide precision and recall summarized across all three classes; to do so, we average precision or recall across each of the three projected classes (decreasing, flat, increasing). The code and data to reproduce all analyses can be found in the public Github repository[69].

**Reporting summary**

Further information on research design is available in the Nature Portfolio Reporting Summary linked to this article.

## Data availability

All data analyzed in the present study can be viewed online at https://covid19scenariomodelinghub.org/ and downloaded at https://github.com/midas-network/covid19-scenario-hub_evaluation/tree/main/data-raw.

## Code availability

All analyses were performed in R, version 4.2.0. All necessary data and code to reproduce analyses is available at https://github.com/midas-network/covid19-scenario-hub_evaluation/ and deposited at https://zenodo.org/record/8415147. For a complete list of packages used, and corresponding versions, see renv.lock file in this repository.

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

## Acknowledgements

The authors thank Matt Ferrari, Ottar Bjørnstad, David Miller, and Tracy Langkilde for their thoughtful comments. LC, JL, JK, JE, and HH were supported by NIGMS 5U24GM132013. EH, KS and RB were supported by NSF RAPID awards DEB-2028301, DEB-2037885, DEB-2126278 and DEB-2220903. EH was also supported by the Eberly College of Science Barbara McClintock Science Achievement Graduate Scholarship in Biology at the Pennsylvania State University. MC, JTD, KM, XX, APyP, and AV were supported by HHS/CDC 6U01IP001137, HHS/CDC 5U01IP0001137 and the Cooperative Agreement no. NU38OT000297 from the Council of State and Territorial Epidemiologists (CSTE). PP, SV, AA, BL, BK, JO, BH, HM, AW, MM, JC, SH, PB, DM acknowledge support from NIH Grant R01GM109718, VDH Grant PV-BII VDH COVID-19 Modeling Program VDH-21-501-0135, NSF Grant No. OAC-1916805, NSF Expeditions in Computing Grant CCF-1918656, NSF RAPID CCF-2142997, NSF RAPID OAC-2027541, US Centers for Disease Control and Prevention 75D30119C05935, DTRA subcontract/ARA S-D00189-15-TO-01-UVA, and UVA strategic funds. Model computation was supported by NSF XSEDE TG-BIO210084 and UVA; and used resources, services, and support from the COVID-19 HPC Consortium (https://covid19-hpc-consortium.org). AB, KB, ML, SJF and LM were supported by CSTE NU38OT000297, CDC Supplement U01IP001136-Suppl, and NIH Supplement R01AI151176-Suppl. ER, JI, MM, and JS were supported by TRACS/NIH grant UL1TR002489; CSTE and CDC cooperative agreement no. NU38OT000297. Funding for the JHU-IDD team was provided by the National Science Foundation (2127976; ST, CPS, JK, ECL, AH), Centers for Disease Control and Prevention (200-2016-91781; ST, CPS, JK, AH, JL, JCL, SLL, CDM, S-mJ), US Department of Health and Human Services/ Department of Homeland Security (ST, CPS, JK, ECL, AH, JL), California Department of Public Health (ST, CPS, JK, ECL, JL), Johns Hopkins University (ST, CPS, JK, ECL, JL), Amazon Web Services (ST, CPS, JK, ECL, AH, JL, JCL), National Institutes of Health (R01GM140564; JL, 5R01AI102939; JCL), and the Swiss National Science Foundation (200021-172578; JCL). Disclaimer. The findings and conclusions in this report are those of the authors and do not necessarily represent the views of the Centers for Disease Control and Prevention. Any use of trade, firm, or product names is for descriptive purposes only and does not imply endorsement by the U.S. Government. This activity was reviewed by CDC and was conducted consistent with applicable federal law and CDC policy (See e.g., 45 C.F.R. part 46, 21 C.F.R. part 56; 42 U.S.C. §241(d); 5 U.S.C. §552a; 44 U.S.C. §3501 et seq.).

## Author contributions

All authors contributed to the generation of COVID-19 Scenario Modeling Hub projections. E.H., J.L., and C.V. conceptualized of the study, developed the methods, evaluated results, and wrote the first draft of the manuscript. E.H. wrote the code and executed analyses.

## Competing interests

J.E. is president of General Biodefense LLC, a private consulting group for public health informatics and has interest in READE.ai, a medical artificial intelligence solutions company. JS and Columbia University disclose partial ownership of SK Analytics. JS discloses consulting for BNI. M.C.R. reports stock ownership in Becton Dickinson & Co., which manufactures medical equipment used in COVID-19 testing, vaccination, and treatment. J.L. has served as an expert witness on cases where the likely length of the pandemic was of issue. The remaining authors declare no competing interests.

## Additional information

Emily Howerton [1] ✉, Lucie Contamin [2], Luke C. Mullany [3], Michelle Qin [4], Nicholas G. Reich[5], Samantha Bents[6], Rebecca K. Borchering [1,7], Sung-mok Jung [8], Sara L. Loo [9], Claire P. Smith[9], John Levander [2], Jessica Kerr[2], J. Espino [2], Willem G. van Panhuis[10], Harry Hochheiser [2], Marta Galanti[11], Teresa Yamana [11], Sen Pei [11], Jeffrey Shaman [11], Kaitlin Rainwater-Lovett [3], Matt Kinsey[3], Kate Tallaksen[3], Shelby Wilson[3], Lauren Shin[3], Joseph C. Lemaitre [8], Joshua Kaminsky[9], Juan Dent Hulse[9], Elizabeth C. Lee [9], Clifton D. McKee[9], Alison Hill [9], Dean Karlen[12], Matteo Chinazzi [13], Jessica T. Davis[13], Kunpeng Mu[13], Xinyue Xiong[13], Ana Pastore y Piontti[13], Alessandro Vespignani [13], Erik T. Rosenstrom[14], Julie S. Ivy[14], Maria E. Mayorga[14], Julie L. Swann[14], Guido España[15], Sean Cavany[15], Sean Moore [15], Alex Perkins [15], Thomas Hladish[16], Alexander Pillai[16], Kok Ben Toh[17], Ira Longini Jr.[16], Shi Chen [18], Rajib Paul[18], Daniel Janies[18], Jean-Claude Thill[18], Anass Bouchnita[19], Kaiming Bi [20], Michael Lachmann[21], Spencer J. Fox [22], Lauren Ancel Meyers[20], Ajitesh Srivastava [23], Przemyslaw Porebski [24], Srini Venkatramanan [24], Aniruddha Adiga[24], Bryan Lewis [24], Brian Klahn [24], Joseph Outten[24], Benjamin Hurt [24], Jiangzhuo Chen[24], Henning Mortveit[24], Amanda Wilson [24], Madhav Marathe[24], Stefan Hoops [24], Parantapa Bhattacharya[24],

Dustin Machi[24], Betsy L. Cadwell[7], Jessica M. Healy [7], Rachel B. Slayton[7], Michael A. Johansson [7], Matthew Biggerstaff [7], Shaun Truelove [9], Michael C. Runge [25], Katriona Shea [1], Cécile Viboud [6,26] ✉ & Justin Lessler [8,9,26] ✉

[1]The Pennsylvania State University, University Park, PA, USA. [2]University of Pittsburgh, Pittsburgh, PA, USA. [3]Johns Hopkins University Applied Physics Lab, Laurel, MD, USA. [4]Harvard University, Cambridge, MA, USA. [5]University of Massachusetts Amherst, Amherst, MA, USA. [6]National Institutes of Health Fogarty International Center, Bethesda, MD, USA. [7]Centers for Disease Control and Prevention, Atlanta, GA, USA. [8]University of North Carolina at Chapel Hill, Chapel Hill, NC, USA. [9]Johns Hopkins University, Baltimore, MD, USA. [10]National Institute of Allergy and Infectious Diseases, Rockville, MD, USA. [11]Columbia University, New York, NY, USA. [12]University of Victoria, Victoria, BC, Canada. [13]Northeastern University, Boston, MA, USA. [14]North Carolina State University, Raleigh, NC, USA. [15]University of Notre Dame, Notre Dame, IN, USA. [16]University of Florida, Gainesville, FL, USA. [17]Northwestern University, Chicago, IL, USA. [18]University of North Carolina at Charlotte, Charlotte, NC, USA. [19]University of Texas at El Paso, El Paso, TX, USA. [20]University of Texas at Austin, Austin, TX, USA. [21]Santa Fe Institute, Santa Fe, NM, USA. [22]University of Georgia, Athens, GA, USA. [23]University of Southern California,  Los Angeles, CA, USA. [24]University of Virginia, Charlottesville, VA, USA. [25]U.S. Geological Survey Eastern Ecological Science Center, Laurel, MD, USA. [26]These authors jointly supervised this work: Cécile Viboud, Justin Lessler. ✉e-mail: ehowerton@psu.edu; viboudc@mail.nih.gov; jlessler@unc.edu

