## [Peer Review File · Nature Communications]

Evaluation of the COVID-19 Scenario Modeling Hub for informing pandemic response under uncertaintyREVIEWER COMMENTS

Reviewer #1 (Remarks to the Author):

This is an important article that reports on a massive attempt to make projections about COVID-19 six months into the future. The authors highlight the value of modelling the course of COVID for policy-making. They also note the challenges that face those who want to get a sense of what the disease's impact on society will look like under different scenarios and well into the future. The Scenario Modeling Hub team engaged in an appropriate ensemble modeling approach to generate predictions, and their work was influenced by the needs, concerns, and focus of public health entities. Their predictions encompassed multiple scenarios over a fairly long period of time. This paper offers a brief overview of the SMH approach and compares the results of its predictions with real-world outcomes.

As is demonstrated in this paper, ensemble models tend to do a better job of making predictions than their component parts. The assumptions and parameters that may lead one model to perform poorly or to have too much confidence in its predictions can be mitigated by bringing several models together. The SMH team relied on a linear opinion pool to aggregate the various models that were used. This methodological approach is not novel, but it is powerful. For comparison, the SMH team examined differences between the ensemble model outcome and a naïve model as well as another ensemble model with a shorter time horizon (4 weeks).

The SMH long-term model performed well when compared to the naïve model and less well when compared with the 4 week ensemble model. The authors identify this outcome as "expected," but should go into more depth here. The 4-week model uses up-to-date data, etc, but the authors should explain why they expected the long run model to fall somewhat short and perhaps consider what the implications are of this result for decision making under uncertainty. Should policy making and planning generally be done on a briefer time scale? Why should we put so much effort into projecting the future under differing scenarios if we are so often going to just do a fairly ok job? The answers to these questions will appear fairly obvious to many in the field and to those involved in this project. They may be far less clear to government officials, political leaders, and the public. Speaking to these audiences is critical, however, if this type of work is to have the impact that it should. The authors note instances in which the SMH long-run ensemble impacted policy-making, and might consider highlighting these moments in this piece and offering more detail.

This work is clearly of significance to the field. It represents the outcome of a large and substantial experiment in our capacity to model a persistent and protean threat to society. It has applications beyond COVID, with such modeling potentially informing planning and policy-making for a variety of diseases. I believe that most readers will wish to hear a little bit more about these potential implications.

The work presented supports the conclusions and claims that are made in the paper. The authors make it clear that there were several conditions under which this approach did not achieve great results. I believe that they should elaborate this theme. Predictive modeling like this is inevitably constrained by our choices of inputs, our assumptions, and the scenarios that we imagine. If a new variant of COVID-19 emerges and makes our predictions substantially less useful, how large of a problem is this? Aren't these the moments that we should be most concerned about and interested in? Periods of stability that can be projected in a fairly straightforward matter seem substantially less compelling as objects of study than those periods in which the model results were confronted with changes in the dominant COVID variant or when the assumptions underpinning the models were challenged. I would be interested to see the authors further highlight the theme of uncertainty (which is clearly at the heart of their research). As a concept, uncertainty proved very challenging for public health leaders to communicate during the pandemic.

The data analysis is sound and the authors are clear about their conclusions, which are grounded in the evidence. The organization of the paper was a bit strange, with a discussion and several conclusions preceding the detailed description of methods. I could see a somewhat more streamlined version of this paper presented in a more standard format with a bit more explicit

discussion of the value of models given what we know in hindsight. This is incredibly impressive work and making sure that it is fully and thoughtfully situated for readers is important.

The methods are consistent with standards in the field, and are described with enough detail for readers to evaluate them and for them. The SMH website also provides extensive information about the teams and models involved.

-Daniel Sledge

Reviewer #2 (Remarks to the Author):

This manuscript describes the process of creating and evaluating models submitted to the COVID-19 Scenario Modeling Hub throughout 16 rounds of forecasts. There are two important aspects of interest: 1) how to retrospectively assess forecasts based on imperfect requests; and 2) how the accuracy of the scenarios in the forecast requests affected the ability of modelers to forecast outcomes of interest.

The discussion lacks sufficient comparison to similar previous efforts, although it mentions new efforts inspired by the SMH. It would be good to address this – were there scenario modeling teams active in public health or other infectious disease communities prior to this pandemic? If so, what methods did they use?

There is no information on how models were elicited (the recruitment method for teams, the instructions provided, and the format of submissions). This would be of interest to future modeling hub efforts. Even a brief mention without redirection to another publication would be helpful.

Specific comments:

Figure 2: there is a typo in the second arrow (scenarios)

Reviewer #3 (Remarks to the Author):

The article reviews the medium to long-term forecast by the US scenario modelling hub. The subject of long-term infectious disease forecast is difficult one, and the article demonstrate very well the value of collaborative effort in tackling it. This work presents a great effort to tackle an unresolved issue of critical importance for public health.

The article presents efforts related to SARS-CoV-2, and clearly delineates (as much as possible) the scope by exploring scenario modelling as opposed to (shorter-term) forecasting.

Very nice way of defining how the scenarios considered did match the reality (bracketing). It seems like broader scenarios would increase the occurrence of bracketing, while restricting the scenario (ideally closer to reality) would make the forecast more accurate (under assumptions of 'correct' modelling/forecast). In that sense, would it be possible to 1) provide more general visualisation of the bracketing (using some kind of standardised axis)? And 2) quantify them in term of perhaps sensitivity/specificity. Broad scenarios would be characterised by high sensitivity and low specificity?

A critical aspect of the scenario model is the validity of the scenarios considered. The model forecast seems to behave relatively well when scenarios are retrospectively judged plausible. Therefore, perhaps more emphasis could be put on the validity of the scenarios modelled? How was the process of defining those? Did it rely on data/analyses? If not, could the authors share more of their experience and perhaps give tentative guidelines as how to establish those?

Eventually, I guess the goal would be that long term scenarios are also based retrospective analyses? We could then see 2 types of models, those that forecast the scenarios and those that forecast the dynamics of transmission for a given scenario?

In figure 3, it is unclear how the SMH projection compared to the ones derived from the 4-week-forecasting hub and/or GAM projections? Should the panel be relative the 4-week-forecasting hub or GAM? Or at least have an equivalent (supplementary) figure(s)? It feels from figure 4 that those perform better, especially when incidence was observed to be increasing? If that's right, this should be discussed? I viewed adding scenarios as relying on more information than in short term forecast. If this is the case, and the added information is sensible/correct, the projection would be expected to improve? Could the authors elaborate on this, perhaps some intuition would be helpful?

Could the author present some (at least preliminary/descriptive, e.g. in SI) results concerning the spatial predictions. Where predictions distributed well spatially? This might become a very important point in term of resource allocations? Similar to the general trend analysis, this could be done in a semi-quantitative way, e.g. defining the regions with high future incidence (or changes in) vs those with low future incidence.

L51: 'A linear opinion pool ensemble of participating models' unclear what this is from reading the abstract, consider re-phrasing to give a better intuition? Similarly in line 100.

L754: check the definition (written form) for precision and recall, I believe some index are wrongly mentioned? I might be missing something, but the denominators are either observed or projected, perhaps call them differently (not N in both cases?).

Figure 1: consider ways to increase font size in pane C.

Figure 3: I assume y-axis on panel C are predictions rounds? Do specify.

SI

Figure S1: a bit more context would read the figure more independently.

Figure S2: y-axis between round 1 and 2 seems different? According to legend they both plot the number of first doses, but the observations seem not consistent?

Fig.26 and some other plot more than 1 predictions for the same time point? Please explain.

Reviewer #1

This is an important article that reports on a massive attempt to make projections about COVID-19 six months into the future. The authors highlight the value of modelling the course of COVID for policy-making. They also note the challenges that face those who want to get a sense of what the disease's impact on society will look like under different scenarios and well into the future. The Scenario Modeling Hub team engaged in an appropriate ensemble modeling approach to generate predictions, and their work was influenced by the needs, concerns, and focus of public health entities. Their predictions encompassed multiple scenarios over a fairly long period of time. This paper offers a brief overview of the SMH approach and compares the results of its predictions with real-world outcomes.

As is demonstrated in this paper, ensemble models tend to do a better job of making predictions than their component parts. The assumptions and parameters that may lead one model to perform poorly or to have too much confidence in its predictions can be mitigated by bringing several models together. The SMH team relied on a linear opinion pool to aggregate the various models that were used. This methodological approach is not novel, but it is powerful. For comparison, the SMH team examined differences between the ensemble model outcome and a naïve model as well as another ensemble model with a shorter time horizon (4 weeks).

The SMH long-term model performed well when compared to the naïve model and less well when compared with the 4 week ensemble model. The authors identify this outcome as "expected," but should go into more depth here. The 4-week model uses up-to-date data, etc, but the authors should explain why they expected the long run model to fall somewhat short and perhaps consider what the implications are of this result for decision making under uncertainty. Should policy making and planning generally be done on a briefer time scale? Why should we put so much effort into projecting the future under differing scenarios if we are so often going to just do a fairly ok job? The answers to these questions will appear fairly obvious to many in the field and to those involved in this project. They may be far less clear to government officials, political leaders, and the public. Speaking to these audiences is critical, however, if this type of work is to have the impact that it should. The authors note instances in which the SMH long-run ensemble impacted policy-making, and might consider highlighting these moments in this piece and offering more detail.

Author response: Thank you for these important questions. We believe scenario projections and forecasts address different goals and both should play a central role in disease planning and response. In our paper, we evaluate scenario projections based on how well these projections match observations after consideration of the plausibility of scenario assumptions. This analysis reflects only one aspect of how scenarios are used, and there are many other ways of evaluating the utility of scenario projections, per the third paragraph of the Discussion. Importantly, there are essential policy questions at longer time horizons that cannot be addressed by forecasts (e.g., optimal timing of vaccine recommendations). In these instances, we believe the scenario projection

approach, which makes specific assumptions about key epidemic drivers, provides crucial insight about future dynamics that is not available otherwise. These scenario projections would not be expected to exactly match observations, especially when scenario assumptions diverge from reality. Further, scenario projections are often used to contrast disease outcomes with and without interventions and anticipate the population benefits of an intervention. Since only one reality will materialize, these projected benefits are difficult to compare to observations as is traditionally done for forecasting, and yet they are particularly useful for quantifying the value of possible interventions.

Our initial expectation was that 4-week ahead forecasts would be closer to observations than scenario projections generated over 3- to 12-month horizons because forecasts are calibrated against more recent data. We also recognize that, on occasion, scenario assumptions that are particularly close to reality could favor projections over forecasts (e.g., with respect to changes in vaccine coverage or NPIs). We believe specifying future drivers will favor scenario projections over forecasts most in cases where those scenario assumptions will affect short-term dynamics and are difficult to predict otherwise.

In our study, short-term forecasts usually outperform scenario projections in how well they match observations, suggesting that recent calibration data is important to exactly capture observations. We also observe rare instances of scenario projections outperforming forecasts (Omicron rounds 11 and 12), which we attribute to particularly accurate scenario assumptions about the properties of an emergent variant. We have made the following changes to better highlight our understanding of, and expectations for, forecasting vs. scenario projections:

1. We have removed the phrase “As expected,” in the fourth paragraph of section **“SMH ensemble consistently outperformed component and comparator models”**.
2. We have added two sentences to the discussion that highlights the complexities of the differences between forecasts and scenario projections: “In addition, despite the fact that forecast models projecting over a shorter time horizon can use more recent information, post-hoc selection of plausible scenario weeks has the potential to “tip the scales” of evaluation in favor of scenario projections, as forecast models are not given the opportunity to project under multiple scenarios. There also remain many open questions about the predictability of infectious disease systems, such as the relative benefits of recent calibration data (which would benefit forecast models) versus knowledge of key drivers of disease dynamics (which would benefit scenario projection models that consider multiple possibilities.”

This work is clearly of significance to the field. It represents the outcome of a large and substantial experiment in our capacity to model a persistent and protean threat to society. It has applications beyond COVID, with such modeling potentially informing planning and policy-making for a variety of diseases. I believe that most readers will wish to hear a little bit more about these potential implications.

Author response: We fully agree that this approach has great potential to inform planning and policy in a wide array of domains. We have added to the last paragraph of Discussion to further highlight this potential:

1. We now include additional examples of multi-model infectious disease scenario projections for other pathogens: “(other notable efforts include multi-model estimation of vaccination impact^{46–48}, planning for future influenza pandemics⁴⁹, and COVID-19 response in South Africa⁵⁰ and the UK⁴⁴)”
2. We also now note the potential of this method to be applied to existing infectious disease concerns in addition to future pandemic threats: “Looking to the future, the lessons learned and the emerging shared hub infrastructure⁵³ can help to provide a more effective, coordinated, and timely response to new pandemic threats and improve mitigation of endemic pathogens.”

The work presented supports the conclusions and claims that are made in the paper. The authors make it clear that there were several conditions under which this approach did not achieve great results. I believe that they should elaborate this theme. Predictive modeling like this is inevitably constrained by our choices of inputs, our assumptions, and the scenarios that we imagine. If a new variant of COVID-19 emerges and makes our predictions substantially less useful, how large of a problem is this? Aren't these the moments that we should be most concerned about and interested in?

Author response: This is a good point, which also connects with our response to the first comment of this reviewer. One of our primary points in the Discussion is that the analyses presented here represent only a start to assessing the *utility* of SMH projections. Accurate projections, as we have measured here (i.e., by comparing projections with observations) are not necessarily useful for planning. For instance, a model may generate accurate projections of new infections, but a decision-maker may only be interested in hospital occupancy metrics. Conversely, useful projections are not necessarily accurate in the sense that they can never be matched to observations (e.g., projections generated by a counterfactual scenario are useful for estimating intervention effects).

SMH projections have addressed variants in a range of different ways, including modeling a new variant that has already emerged elsewhere, modeling a hypothetical variant, and modeling the “mix” of variants that has been characteristic in the post-Omicron era. Each of these kinds of projections have been used by public health officials. Further, we have found that across 16 rounds of SMH projections, it took on average 22 weeks for scenarios to be invalidated by the emergence of a new SARS-CoV-2 variant. This indicates that even in periods of rapid viral changes, scenario projections can remain valid over a time horizon that is actionable. Lastly, the rounds that achieved the best performance statistics were rounds 11 and 12 which addressed the emergence of the Omicron variant. Hence, we do not see a systematic pattern in which projections that address the emergence of new variants would be inherently less useful.

We have added more detail to the second paragraph of the Discussion to better address how variants have been included in SMH projections.

1. First, we distinguish between the emergence of *unanticipated* variants that invalidated scenario assumptions, and the ways in which SMH has addressed *known* variants: “While the emergence of unanticipated variants presented a challenge to long-term projections, SMH often showed strength in an ability to anticipate the impact of new variants that were emerging elsewhere in the world.”
2. Later in this paragraph, we added discussion of the emergence of unanticipated variants as an important uncertainty that could affect important policy questions: “Notably, SMH projections also provided key information to guide policy recommendations by allowing us to compare different intervention strategies while simultaneously accounting for major uncertainties, such as modeling the emergence of a hypothetical variant.”

Periods of stability that can be projected in a fairly straightforward matter seem substantially less compelling as objects of study than those periods in which the model results were confronted with changes in the dominant COVID variant or when the assumptions underpinning the models were challenged. I would be interested to see the authors further highlight the theme of uncertainty (which is clearly at the heart of their research). As a concept, uncertainty proved very challenging for public health leaders to communicate during the pandemic.

Author response: We appreciate this comment and agree that managing uncertainty is an important theme in our work. We have added a clear statement of this in the final paragraph of the discussion: “The SMH process, which uses the power of multi-model ensembles and strategic selection of future scenarios to manage uncertainty¹⁵, has already been replicated in other settings⁴⁸ and for other pathogens⁴⁹.”

The data analysis is sound and the authors are clear about their conclusions, which are grounded in the evidence. The organization of the paper was a bit strange, with a discussion and several conclusions preceding the detailed description of methods. I could see a somewhat more streamlined version of this paper presented in a more standard format with a bit more explicit discussion of the value of models given what we know in hindsight. This is incredibly impressive work and making sure that it is fully and thoughtfully situated for readers is important.

Author response: Thank you. We have written this manuscript for a general science audience following the standard format for Nature Communications. We believe this format is well suited for our manuscript, as technical details could obscure our broader conclusions, but we defer to the judgement of the editor on the best structure for our manuscript.

The methods are consistent with standards in the field, and are described with enough detail for readers to evaluate them and for them. The SMH website also provides extensive information about the teams and models involved.

-Daniel Sledge

Reviewer #2

This manuscript describes the process of creating and evaluating models submitted to the COVID-19 Scenario Modeling Hub throughout 16 rounds of forecasts. There are two important aspects of interest: 1) how to retrospectively assess forecasts based on imperfect requests; and 2) how the accuracy of the scenarios in the forecast requests affected the ability of modelers to forecast outcomes of interest.

The discussion lacks sufficient comparison to similar previous efforts, although it mentions new efforts inspired by the SMH. It would be good to address this – were there scenario modeling teams active in public health or other infectious disease communities prior to this pandemic? If so, what methods did they use?

Author response: This is a good point. We have added information about prior efforts in two ways:

1. In addition to the existing references to prior multi-model forecasting efforts (references 9-13) and previous literature on how to integrate multiple models with scenario projections (14-16), we have added additional examples of prior multi-model scenario projection efforts to the discussion: (other notable efforts include multi-model estimation of vaccination impact⁴⁶⁻⁴⁸, planning for future influenza pandemics⁴⁹, and COVID-19 response in South Africa⁵⁰ and the UK⁴⁴). This includes a model comparison study of vaccination effects for Dengue (reference 47) and Rotavirus (reference 48) and pandemic influenza interventions (reference 49), as well as other examples of multi-model consortiums for COVID-19 response (references 44,50).
2. We have added a sentence to the second paragraph of Methods section “*Elicitation methods and models submitting projections to SMH*” that discusses the wide variety of prior experience across modeling teams: “Of participating models, prior experience in public health modeling varied substantially, ranging from teams with newly built models to address the COVID-19 pandemic and those with long-established relationships with local, state, and national public health agencies.”

There is no information on how models were elicited (the recruitment method for teams, the instructions provided, and the format of submissions). This would be of interest to future modeling hub efforts. Even a brief mention without redirection to another publication would be helpful.

Author response: Thank you for this suggestion, we agree this is useful information to include. We have made the following changes to incorporate this information into the manuscript:

1. We have modified a sentence in the fifth paragraph of the Introduction to include the open nature of SMH calls for projections: “Open calls for projections have

yielded participation from thirteen teams overall, with some teams providing projections only for certain rounds or states.”

2. We added an additional paragraph to the Methods to provide more details about model elicitation. We renamed the subsection accordingly to “Elicitation methods and models submitting projections to SMH”. In this paragraph, we discuss SMH elicitation strategies, and we cite Loo et al., a manuscript currently in revision, that details the philosophy and processes of SMH. We anticipate adding the full citation for this reference in the proof stage of publication.

Specific comments:

Figure 2: there is a typo in the second arrow (scenarios)

Author response: Thank you for catching this typo. We have corrected it.

Reviewer #3

The article reviews the medium to long-term forecast by the US scenario modelling hub. The subject of long-term infectious disease forecast is difficult one, and the article demonstrate very well the value of collaborative effort in tackling it. This work presents a great effort to tackle an unresolved issue of critical importance for public health.

The article presents efforts related to SARS-CoV-2, and clearly delineates (as much as possible) the scope by exploring scenario modelling as opposed to (shorter-term) forecasting.

Very nice way of defining how the scenarios considered did match the reality (bracketing). It seems like broader scenarios would increase the occurrence of bracketing, while restricting the scenario (ideally closer to reality) would make the forecast more accurate (under assumptions of ‘correct’ modelling/forecast). In that sense, would it be possible to 1) provide more general visualisation of the bracketing (using some kind of standardised axis)? And 2) quantify them in term of perhaps sensitivity/specificity. Broad scenarios would be characterised by high sensitivity and low specificity?

Author response: Thank you for this thoughtful suggestion. We think such a figure would be very useful; however, there are issues that limit our ability to apply these ideas in practice. First, our scenarios are typically based on the interaction between 2 main epidemic drivers (e.g., variant transmissibility, vaccination uptake, NPIs, etc.), and it is difficult to find a way to translate these assumptions onto a standard axis. Each type of axis has a very different scale and interpretation.

We believe Figures S2-S6 allow the comparison you are discussing for singular scenario axes. For example, Round 2 scenarios (as shown in Figure S2) were the broadest and captured the observations, whereas Round 4 scenarios were closer together and did not capture observations. We have added a sentence to the caption of

Table 1 to direct readers to these figures: “Visualization of scenario assumptions and bracketing are provided in Figures S2-S6.”

To visualize the two-dimensional nature of these scenarios, one could keep each scenario assumption on a separate axis (that retains the appropriate scale). There are only a few SMH rounds where we can evaluate both axes (Rounds 6, 7, 11, 12), and unfortunately these were two sets of “repeat rounds”, such that scenario specifications were intentionally similar across Rounds 6 and 7, and 11 and 12. We have created an example for Rounds 6 and 7 and have included it as a second panel in Figure S3.

We are not sure we fully follow the reviewer in their suggestion to use sensitivity and specificity for scenario assumptions, but we think it is helpful to think about how closely scenarios bracket an observation versus the likelihood scenarios will bracket. We have recently written a manuscript on the theory and process of scenario design which discusses such considerations (see our response to your next point).

A critical aspect of the scenario model is the validity of the scenarios considered. The model forecast seems to behave relatively well when scenarios are retrospectively judged plausible. Therefore, perhaps more emphasis could be put on the validity of the scenarios modelled? How was the process of defining those? Did it rely on data/analyses? If not, could the authors share more of their experience and perhaps give tentative guidelines as how to establish those? Eventually, I guess the goal would be that long term scenarios are also based retrospective analyses? We could then see 2 types of models, those that forecast the scenarios and those that forecast the dynamics of transmission for a given scenario?

Author response: These are very important questions. The goal of section “*SMH scenarios usually bracketed future epidemic drivers*” is to discuss the post-hoc validity of SMH scenarios. SMH scenarios were informed by available data whenever possible, which we have clarified in the first paragraph of this section: “Typically, these levels aimed to bracket the future values of important epidemic drivers using information available at the time of scenario design...”. The reason we take a scenario approach is because we do not believe future epidemic drivers can be forecasted accurately enough to generate useful predictions. Thus, the choice of scenarios is an extremely important topic, as you note, and we believe scenario design should be primarily driven by the intended use(s) of the scenarios and the major drivers of uncertainty at the time of projections.

We have recently written a manuscript on the theory and process of scenario design, which is currently under peer review (Runge et al.); we will add the full reference at the proof stage if accepted in time. To better highlight this discussion, we have changed the name of the relevant methods section to “*Scenario design and plausibility*” and we have added a sentence in that section to highlight the importance of clearly-defined purposes in scenario design and a good grasp of major sources of uncertainties: “Scenario design was guided by one or more primary purposes⁵⁶, which were often informed by public health partners and our hypotheses about the most important uncertainties at the time.”

In figure 3, it is unclear how the SMH projection compared to the ones derived from the 4-week-forecasting hub and/or GAM projections? Should the panel be relative the 4-week-forecasting hub or GAM? Or at least have an equivalent (supplementary) figure(s)? It feels from figure 4 that those perform better, especially when incidence was observed to be increasing? If that’s right, this should be discussed? I viewed adding scenarios as relying on more information than in short term forecast. If this is the case, and the added information is sensible/correct, the projection would be expected to improve? Could the authors elaborate on this, perhaps some intuition would be helpful?

Author response: Thank you for these questions. In Figure 3, we plot the 95% coverage (panel A) and average normalized WIS (panel B) for the 4-week forecast model using a gray line. The purpose of panel C is to compare the SMH ensemble to individual models, rather than the ensemble to a null alternative. We do, however, provide results comparing the SMH ensemble to each null comparator in the supplement. We now reference this supplementary figure in the caption of Figure 3: “See Figure S46-Figure S47 for 50% and 95% coverage of all targets and see Figure S22 for comparison of WIS for SMH ensemble to each null comparator.”

Figure 4 also shows results for null alternatives, including the 4-week forecast model, a model that assumes trends continue, and what we would expect if classifications were made at random. These results, including SMH performance relative to these alternatives, are discussed in the second paragraph of section “**While adding value over comparators, SMH projections struggled to anticipate changing disease trends**”.

You make a good point that scenarios may give additional information that forecasts may not have. We have added a sentence to the discussion accordingly: “In addition, despite the fact that forecast models projecting over a shorter time horizon can use more recent information, post-hoc selection of plausible scenario weeks has the potential to “tip the scales” of evaluation in favor of scenario projections, as forecast models are not given the opportunity to predict under multiple scenarios.”

Could the author present some (at least preliminary/descriptive, e.g. in SI) results concerning the spatial predictions. Where predictions distributed well spatially? This might become a very important point in term of resource allocations? Similar to the general trend analysis, this could be done in a semi-quantitative way, e.g. defining the regions with high future incidence (or changes in) vs those with low future incidence.

Author response: This is a very good question, and we believe it warrants thorough investigation. We have provided a figure in the supplement to assess the validity of

projections by geographic region and population size (Figure S57), in addition to figures that show state-specific trends for projections and observations (Figures S24-S33). In other papers reporting SMH results (references 5 and 22), we assessed the ability of SMH projections to predict the relative rank of cumulative incidence across states (i.e., do SMH projections identify the states that will have many cases, hospitalizations, or deaths, and correspondingly higher need for resources?). These published analyses focused only on specific rounds and specific projection horizons; we believe it is important to broaden these analyses to all rounds and horizons. Given the scope of this manuscript is already quite broad and focused on incident projections, we believe such an analysis would be better suited for a separate manuscript. However, we have made the following changes to highlight the value of spatial analyses and stress the need for more in-depth analyses in the future.

1. We have updated Figure S57, which plots projection performance of each state by the population size of that state. Now each state abbreviation is colored by the US census region of that state. There are no clear regional patterns, and we find a weak relationship between projection performance and state size. Also, we have added to the caption of Figure S57 a reference to the papers presenting the analyses described above: “Other SMH analyses have assessed the ability of SMH projections to accurately predict rank of states according to cumulative targets for specific rounds and weeks^{20,21}. “
2. Importantly, we have added the idea of spatial resource allocation to the Discussion (third paragraph); the sentence now reads: “Alternatively, one might use the full set of scenarios to allocate resources or inform response plans to potential surges in disease incidence; in this case, we might evaluate how well SMH projections identified states with highest need or the extent to which planning around extremes from pessimistic projections would have led to over- or under-allocation of resources.”

L51: ‘A linear opinion pool ensemble of participating models’ unclear what this is from reading the abstract, consider re-phrasing to give a better intuition? Similarly in line 100. **Author response:** Thank you for noting this. We have changed the text in the abstract to read “An ensemble of participating models that preserved variation between models (using the linear opinion pool method)” and the introduction to read “Projections were aggregated using the linear opinion pool method¹⁸, which preserves variation between model projections¹⁹.”

L754: check the definition (written form) for precision and recall, I believe some index are wrongly mentioned? I might be missing something, but the denominators are either observed or projected, perhaps call them differently (not N in both cases?).

Author response: We agree the denominators should be number observed or projected of each class, and our presentation of these formulas could be clearer. We have added a table that denotes N_{p_o} for each combination of projected and observed classes and included the notation in our examples. We also now include a note about summarizing these metrics across classes: “In some instances, we provide precision

and recall summarized across all three classes; to do so, we average precision or recall across each of the three projected classes (decreasing, flat, increasing).”

Figure 1: consider ways to increase font size in pane C.

Author response: This is a useful suggestion; we have consolidated some of the language in the table which allowed us to increase the font size.

Figure 3: I assume y-axis on panel C are predictions rounds? Do specify.

Author response: Thank you for this suggestion, we have made this change.

SI

Figure S1: a bit more context would read the figure more independently.

Author response: We have added more information to the caption of Figure S1 to provide important context: “Figure S1: Number of weeks projected versus number of weeks for which scenario specifications were valid across SMH rounds. (A) Depending on the needs and context of each round, SMH projections were made for 12 weeks (Rounds 11, 12), 26 weeks (Rounds 1-7, 9, and 16), 40 weeks (Round 15), or 52 weeks (Rounds 13, 14) (gray dashed line), Retrospectively, we assessed how many weeks into the future scenario specifications were valid (“effective projection horizon”, black line). We assumed scenario specifications were invalidated when a SARS-CoV-2 variant emerged that was not included in the scenario specifications. (B) Histogram of effective projection horizon across 14 public rounds, including median and mean effective horizon (dotted vertical lines).”

Figure S2: y-axis beyween round 1 and 2 seems different? According to legend they both plot the number of first doses, but the observations seem not consistent?

Author response: We appreciate you catching this error on our part. Round 1 specified distributed doses, whereas Rounds 2 and 3 specified administered first doses. We have made this correction in the figure caption.

Fig.26 and some other plot more than 1 predictions for the same time point? Please explain.

Author response: We have modified the caption here and in Fig S27 to more clearly explain why there are two predictions for each week: “Note, the scale of the y-axis is not consistent across panels and for this round there are two most plausible scenarios because NPI scenario specifications could not be validated.”

REVIEWERS' COMMENTS

Reviewer #1 (Remarks to the Author):

I have read the revised manuscript and the responses to the reviewers. I am satisfied with the revisions. I would be interested to read a future piece that reflects on the experience of putting all of this together and attempting to inform the public and policymakers.

Reviewer #3 (Remarks to the Author):

I am satisfied with the authors response, and I am happy to recommend acceptance of the manuscript. I do think the manuscript addresses a very important issue and does so in critical way. It will therefore positively contribute to future effort and discussion surrounding medium to long-term forecasting of infectious diseases.

While a critical aspect of long-term forecasting, I do understand that a long discussion surrounding the design of scenarios would be outside the scope of this paper. However, I strongly recommend that a reference to the theoretical paper mentioned by the authors in their response (on the theory and process of scenario design, which is currently under peer review (Runge et al.)) is included, even if not yet published (e.g. as a pre-print reference?). I look forward reading the additional paper.

Reviewer #1 (Remarks to the Author):

I have read the revised manuscript and the responses to the reviewers. I am satisfied with the revisions. I would be interested to read a future piece that reflects on the experience of putting all of this together and attempting to inform the public and policymakers.

Author response: Thank you for the helpful suggestions.

Reviewer #3 (Remarks to the Author):

I am satisfied with the authors response, and I am happy to recommend acceptance of the manuscript. I do think the manuscript addresses a very important issue and does so in critical way. It will therefore positively contribute to future effort and discussion surrounding medium to long-term forecasting of infectious diseases.

While a critical aspect of long-term forecasting, I do understand that a long discussion surrounding the design of scenarios would be outside the scope of this paper. However, I strongly recommend that a reference to the theoretical paper mentioned by the authors in their response (on the theory and process of scenario design, which is currently under peer review (Runge et al.)) is included, even if not yet published (e.g. as a pre-print reference?). I look forward reading the additional paper.

Author response: Thank you for your thoughtful comments. We agree Runge *et al.* is an essential reference for this paper. As such, we have posted this manuscript as a pre-print on medRxiv so it can be referenced.